# Structure of putative tumor suppressor ALDH1L1

Yaroslav Tsybovsky [1✉], Valentin Sereda[2], Marcin Golczak[3], Natalia I. Krupenko [2,4] &
Sergey A. Krupenko [2,4✉]

Putative tumor suppressor ALDH1L1, the product of natural fusion of three unrelated genes, regulates folate metabolism by catalyzing $NADP^+$-dependent conversion of 10-formyltetrahydrofolate to tetrahydrofolate and $CO_2$. Cryo-EM structures of tetrameric rat ALDH1L1 revealed the architecture and functional domain interactions of this complex enzyme. Highly mobile N-terminal domains, which remove formyl from 10-formyltetrahydrofolate, undergo multiple transient inter-domain interactions. The C-terminal aldehyde dehydrogenase domains, which convert formyl to $CO_2$, form unusually large interfaces with the intermediate domains, homologs of acyl/peptidyl carrier proteins (A/PCPs), which transfer the formyl group between the catalytic domains. The 4'-phospho-pantetheine arm of the intermediate domain is fully extended and reaches deep into the catalytic pocket of the C-terminal domain. Remarkably, the tetrameric state of ALDH1L1 is indispensable for catalysis because the intermediate domain transfers formyl between the catalytic domains of different protomers. These findings emphasize the versatility of A/PCPs in complex, highly dynamic enzymatic systems.

[1] Cancer Research Technology Program, Leidos Biomedical Research Inc., Frederick National Laboratory for Cancer Research, 8560 Progress Drive, Frederick, MD 21701, USA. [2] Nutrition Research Institute, University of North Carolina at Chapel Hill, 500 Laureate Way, Kannapolis, NC 28081, USA. [3] Department of Pharmacology, School of Medicine, Case Western Reserve University, 10900 Euclid Ave, Cleveland, OH 44106, USA. [4] Department of Nutrition, University of North Carolina at Chapel Hill, 135 Dauer Drive, Chapel Hill, NC 27599, USA. ✉email: Yaroslav.Tsybovsky@nih.gov; sergey_krupenko@unc.edu

ALDH1L1 (10-formyltetrahydrofolate dehydrogenase), an enzyme of folate metabolism, regulates the availability of one-carbon groups for folate-dependent biochemical reactions[1]. The importance of this regulation is emphasized by high abundance of the enzyme in the liver, the main organ of folate metabolism, as well as by tight control of the ALDH1L1 expression during embryonic development and by the role of the protein as a pan-astrocyte marker[1–3]. The regulatory role of ALDH1L1 is linked to its catalytic reaction, the NADP+-dependent conversion of 10-formyltetrahydrofolate (10-fTHF) to tetrahydrofolate (THF) and $CO_2$, which irreversibly removes one-carbon groups from the folate pool, thus diminishing the anabolic capacity[1,4] (Fig. 1a). It has been proposed that this reaction interferes with rapid cellular proliferation, but at the same time supports homeostasis in non-proliferating cells by supplying THF for the reaction of conversion of serine to glycine and for the formate metabolism[1,4,5]. The role of the enzyme in supporting glycine production has been recently demonstrated in the *Aldh1l1* knockout mouse model, with livers of ALDH1L1-deficient mice having decreased levels of THF, glycine and glycine conjugates[5]. Lately, the enzyme's function has been linked to NADPH production and oxidative stress[6]. ALDH1L1 is also considered a putative tumor suppressor[4]. This role is supported by findings that the protein is strongly and ubiquitously downregulated in malignant tumors and cancer cell lines[7,8], the effect associated with hypermethylation of the *ALDH1L1* promoter[9–13]. Of note, expression of ALDH1L1 in cancer cell lines produces strong antiproliferative effects by activating specific apoptotic pathways[7,14–18]. In further support of the suppressive effect of ALDH1L1 on proliferation, the enzyme is strongly downregulated in S-phase of the cell cycle through proteasomal degradation but is elevated in quiescent cells[19]. Although the knockout of *Aldh1l1* in mice did not cause the initiation of malignant lesions, it promoted the growth of larger liver tumors initiated by a chemical carcinogen[20].

*ALDH1L1* originated from a natural fusion of three unrelated genes, the phenomenon defining the structural organization of the ALDH1L1 enzyme[1]. The protein exists as a homotetramer, with each 902 amino acid-long protomer organized in three distinct functional domains (Fig. 1b, c). The N-terminal domain ($N_t$, aa 1-310) carries the folate-binding site and has sequence and structural similarity to methionyl-tRNA$^{Met}$-formyltransferase (FMT), the enzyme involved in translation initiation in mitochondria[21]. FMT formylates the initiator Met-tRNA$^{Met}$ by transferring the formyl group from 10-fTHF, thus using the same substrate as ALDH1L1[22–24]. The C-terminal domain ($C_t$, aa 405–902) belongs to the family of aldehyde dehydrogenases (ALDHs), the group of enzymes catalyzing the conversion of a large variety of aldehydes to corresponding acids using NAD+ or NADP+ as the electron acceptor[25]. The $C_t$ domain shares up to 50% sequence similarity with members of this family and has a typical ALDH fold, which includes NAD(P)+-binding, catalytic, and oligomerization sub-domains[26,27]. The $C_t$ domain contains all critical catalytic residues conserved in ALDHs, including Cys707, which plays the role of the catalytic center nucleophile[27,28]. Accordingly, the $C_t$ domain catalyzes the conversion of short-chain aldehydes to corresponding acids in vitro, but it is not known whether ALDH1L1 participates in aldehyde oxidation in vivo[1]. Finally, the intermediate domain (Int, aa 314–397) linking the $N_t$ and $C_t$ domains is a homolog of a group of small, structurally closely related carrier proteins involved in fatty acid, polyketide, and non-ribosomal peptide biosynthesis[29]. A characteristic feature of these acyl/peptidyl carrier proteins (A/PCPs) is the 4′-phosphopantetheine prosthetic group (4′-PP) covalently attached to a serine residue through a phosphoester bond[30,31]. This prosthetic group serves

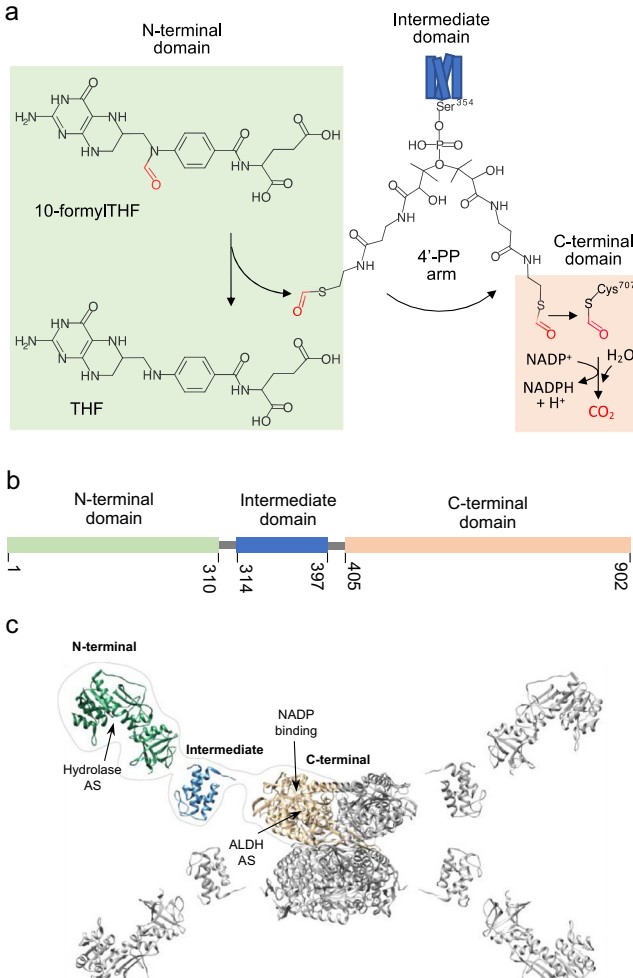

**Fig. 1 Domain organization and catalytic function of 10-formyltetrahydrofolate dehydrogenase (ALDH1L1). a** ALDH1L1 converts 10-formyltetrahydrofolate (10-formylTHF) to tetrahydrofolate (THF) and $CO_2$ in two sequential enzymatic reactions, the cleavage of the formyl group in the folate-binding N-terminal domain and the NADP+-dependent oxidation of the formyl to carbon dioxide in the aldehyde dehydrogenase C-terminal domain. The intermediate domain, which is homologous of acyl/peptidyl carrier proteins, transports the formyl group, attached to the 4′-phosphopantetheine prosthetic arm (4′-PP), from the N-terminal to the C-terminal domain. **b** Primary structure of ALDH1L1 with indicated domain boundaries. **c** Schematic of spatial organization of the ALDH1L1 tetramer based on available atomic structures of its individual domains. PDB structures 1s3i, 2cq8, and 2o2p were used for the N-terminal, intermediate, and C-terminal domains, respectively. For one protomer the domains are labeled and colored as in **b**. Arrows mark the positions of the hydrolase and aldehyde dehydrogenase active sites (AS) and the NADP+ binding site.

as a flexible arm enabling the transfer of building blocks between subunits of multi-enzyme complexes[30,31].

Functional studies of ALDH1L1, its numerous mutants and engineered constructs together with structural and functional characterization of the individual domains provided insight into the enzyme catalytic machinery[21,26–29,32–41]. Overall, in the ALDH1L1 catalysis, the 4′-PP arm of the Int domain transfers the formyl group cleaved from 10-fTHF in the folate-binding $N_t$ domain to the $C_t$ domain, where it is oxidized to carbon dioxide[1] (Fig. 1a). To execute this mechanism, in addition to the flexibility of the 4′-PP moving arm, sufficient mobility of ALDH1L1 domains relative to each other is necessary. This complicates the

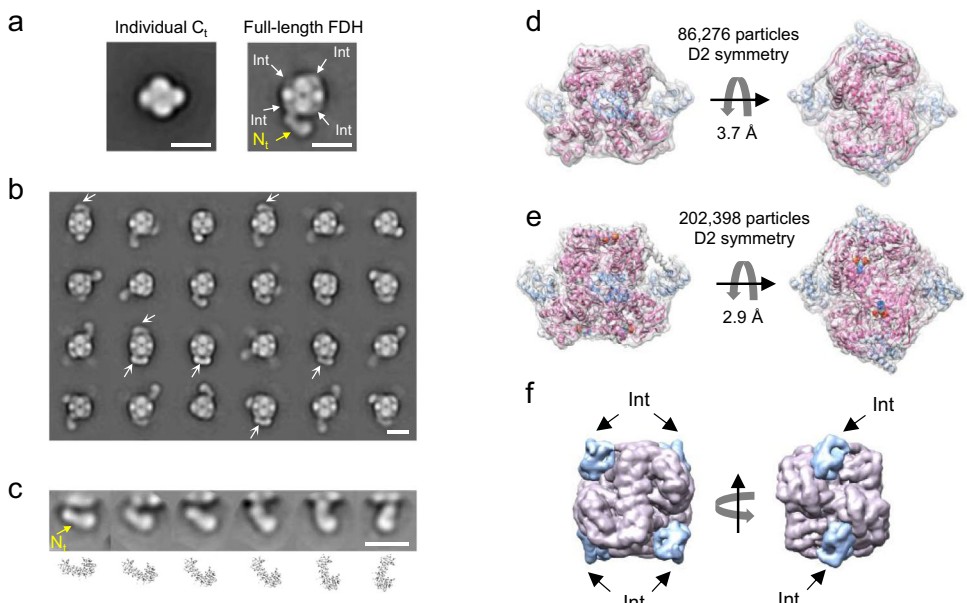

**Fig. 2 The architecture of ALDH1L1 includes a rigid core and highly mobile N-terminal domains. a** Comparison of negative-stain EM 2D class averages of the individually expressed tetrameric C-terminal domain ($C_t$, left panel) and the full-length protein (right panel) was used for identification of intermediate domains (Int) adjacent to the $C_t$ domain within each protomer. Only a single N-terminal domain ($N_t$) is clearly visible. **b** Selected negative-stain 2D classes illustrate the range of positions assumed by the $N_t$ domains within the ALDH1L1 tetramer. Arrows point to the position of the $N_t$ domain in which it closely interacts with the $C_t$ domain. **c** The range of motion of the $N_t$ domain illustrated by a selection of negative-stain 2D classes. Below: cartoon representations of the crystal structure of $N_t$ in the corresponding orientations. **d, e** Cryo-EM structures of ligand-free ALDH1L1 (**d**) and ALDH1L1 in complex with NADP$^+$ (**e**). The maps are displayed as transparent surfaces. Atomic models are shown in cartoon representation, with Int and $C_t$ domains colored light-blue and pink, respectively. Bound NADP$^+$ molecules are shown in sphere representation. **f** Cryo-EM structure of the rigid core of ALDH1L1 corresponds well to the negative-stain EM 2D classes. Results of segmentation of the ligand-free map are shown. Scale bars in panels **a**–**c** correspond to 10 nm.

structural analysis of the full-length ALDH1L1 protein. Indeed, while several crystal structures of $N_t$ and $C_t$ domains were reported[21,27,38–41], and an NMR structure of a synthetic Int domain devoid of the 4′-PP arm is available (PDB 2cq8), the structure of the full-length enzyme has not been resolved so far. Here, we report the structures of ligand-free and NADP$^+$-bound full-length ALDH1L1 at resolutions of 3.7 Å and 2.9 Å, respectively, obtained by cryo-electron microscopy (cryo-EM). This study provides insights into the ALDH1L1 structure and function by (i) demonstrating high mobility of the $N_t$ domains, which form transient complexes with other structural units; (ii) describing the unusual mode of interaction between the Int and $C_t$ domains, with a large contact interface atypical for A/PCPs, and (iii) revealing unique pairing of the Int and $C_t$ domains, which requires the tetrameric organization for catalysis.

## Results

**Overall architecture of full-length ALDH1L1.** To reveal the domain organization of ALDH1L1 we performed negative-stain EM (NS-EM) of the full-length ligand-free protein (*Rattus norvegicus* ALDH1L1 produced in insect cells using a baculovirus expression system) as well as of its individually expressed $C_t$ domain, which forms the rigid tetrameric core of the full-length enzyme. The $C_t$ core was clearly visible in the 2D class averages (Fig. 2a, b), enabling the identification of all four Int domains. All Int domains in full-length ALDH1L1 sat closely to the C-terminal core and were arranged in an apparently symmetrical manner. The high contrast provided by negative staining also allowed, in many cases, to resolve the $N_t$ domains both in raw micrographs (Supplementary Fig. 1) and 2D class averages (Fig. 2b). The $N_t$ domains assumed variable positions with respect to the rest of the protein, indicating their high mobility. Focused 2D classification revealed that the $N_t$ domain sampled the entire range of positions

between the one oriented away from the protein central core and the one in contact with the $C_t$ domain (Fig. 2c). We did not observe any obvious coordination in positioning of the four $N_t$ domains in the ALDH1L1 tetramer in NS-EM experiments, a finding suggesting asynchronous movement of these domains in the full-length protein. However, we noticed that a conformation in which the $N_t$ domain is tightly packed against the $C_t$ core was encountered repeatedly (Fig. 2b). The high mobility of the $N_t$ domain illustrated by the NS-EM data is likely required for the multi-step catalytic mechanism of the enzyme.

Subsequently, we used cryo-EM to characterize the structure of ligand-free ALDH1L1 at high resolution (Fig. 2d). A 3.7-Å map was obtained from a dataset of 86,276 particles when D2 symmetry was imposed. The rigid $C_t$ core and four Int domains were clearly resolved in this structure, while the highly mobile $N_t$ domains were not visible in the symmetrical map. To elucidate the potential effect of NADP$^+$ on the structural organization of ALDH1L1, we also prepared cryo-EM grids after adding 1 mM NADP$^+$ to the full-length protein. Single-particle analysis of the new dataset containing 202,398 particles produced a 2.9-Å resolution map (Fig. 2e). The arrangement of the $C_t$ core and Int domains in this structure was identical to that in the ligand-free protein and corresponded well to the configuration of the $C_t$ and Int modules revealed by NS-EM (Fig. 2f).

**Structures of the Int and $C_t$ domains.** Both the Int and $C_t$ domains were clearly resolved in the cryo-EM maps (Fig. 3), which permitted the building of atomic models for ligand-free and NADP$^+$-bound ALDH1L1. The entire 4′-PP prosthetic group was also well defined in the cryo-EM density (Fig. 3a). As expected based on a previous study[29], the Int domain exhibits a fold typical of A/PCPs (Fig. 3a, b)[42]. Accordingly, its structure consists of three major α-helices (I, II, and IV) forming a loose

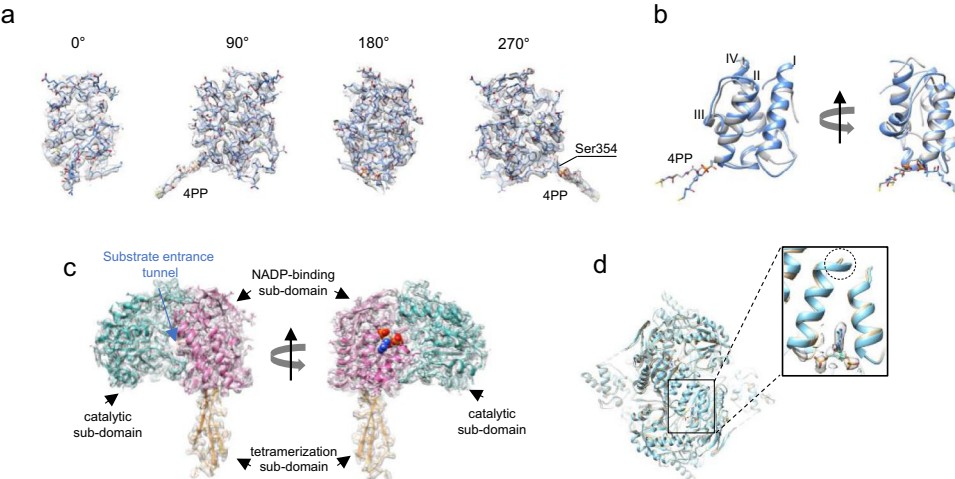

**Fig. 3 Structures of the intermediate and $C_t$ domains of ALDH1L1. a** Angular views of the intermediate domain. The molecular model is shown as *sticks*, and the cryo-EM density is represented by a semi-transparent surface. 4PP: 4'-phosphopantetheine. **b** Superposition of the intermediate domain of ALDH1L1 (*light blue*) and the carrier domain of holo-non-ribosomal peptide synthetase (PDB 4zxh[84], *gray*). The alpha-helices are labeled I to IV. The r.m.s.d. between the two structures is 1.98 Å. **c** Structure of the protomer of the C-terminal domain. The three sub-domains are colored individually and labeled. Bound NADP$^+$ is shown in *sphere* representation. **d** Superposition of the $C_t$ domains of ligand free (*gray*) and NADP$^+$-bound (*light blue*) ALDH1L1. The binding of NADP$^+$ induces local structural changes in one helix forming the binding cleft for the adenine moiety (*dashed* circle). Cryo-EM density for NADP$^+$ is shown as semi-transparent surface.

bundle, with another short helix (III) bridging helices II and IV. Helices I and II are connected by a long 19-residue linker that forms a loop and a helical turn. Serine 354, the site of the 4'-PP group attachment, is located in the beginning of helix II.

Each $C_t$ domain contains the catalytic, NADP$^+$-binding and oligomerization sub-domains (Fig. 3c). The deep substrate entrance tunnel is located between the catalytic and NADP$^+$-binding sub-domains and leads to the catalytic cysteine 707. The NADP$^+$ binding site is situated on the side opposite to the substrate entrance tunnel. In the cryo-EM maps of ALDH1L1, the structure of the $C_t$ domain was very similar to the crystal structures of individual $C_t$ domains previously reported[27]. The root mean square deviation (r.m.s.d.) between the protomer of the $C_t$ domain solved by cryo-EM and the protomer of the corresponding X-ray structure (PDB 2o2p or 2o2q[27]) was 0.46 Å and 0.75 Å, respectively, for ligand-free and NADP$^+$-bound proteins. The r.m.s.d. between the entire tetramers was slightly higher (0.621 Å and 1.08 Å, respectively), suggesting slight differences in protomer positions in full-length ALDH1L1, which may be a consequence of the interactions between the $C_t$ and Int domains. It is also possible that the conformation of the protein in the crystal structures was affected by crystal packing.

We found that the cryo-EM structures of ligand-free and NADP$^+$-bound ALDH1L1 were highly similar (r.m.s.d. = 0.86 Å) (Fig. 3d). A small difference was observed in the NADP$^+$-binding site, where the C-terminal end of the helix formed by residues 652–666 was better ordered in the presence of the coenzyme. Only the AMP portion of bound NADP$^+$ was clearly visible in the cryo-EM map, while there was no density for the rest of the cofactor. Of note, weak density for the nicotinamide riboside of NAD$^+$ and NADP$^+$ is commonly observed in aldehyde dehydrogenases[27,43,44].

**Unique pairing of Int and $C_t$ domains**. In both ligand-free and NADP$^+$-bound ALDH1L1, each Int domain formed contacts primarily with one of the four $C_t$ domains (Fig. 4a). The Int and $C_t$ domains of the same protomer were separated by a 20-Å-long extended linker consisting of residues 397–404 (Fig. 4b) and did not interact with each other. Instead, the Int domain of chain A

was docked at the substrate entrance tunnel of the $C_t$ domain of chain C, whereas the Int domain of chain C was paired to the $C_t$ domain of chain A (Fig. 4c, d). An identical arrangement was found for protomers B and D. Of note, the tetrameric $C_t$ core of ALDH1L1 is composed of two homodimers formed by protomers A/B (dimer 1) and C/D (dimer 2). Therefore, each Int domain of ALDH1L1 is paired to a $C_t$ domain of the opposite dimer (Fig. 4d). Out of the eight residues composing the linker in rat ALDH1L1, five are negatively charged amino acids, which makes the linker highly hydrophilic. Sequence alignments showed that this property of the linker is preserved across species as well as between ALDH1L1 and ALDH1L2 proteins (Fig. 4b), suggesting that hydrophilicity and negative charge of the linker are important for the function of the enzyme. Of note, although there was continuous cryo-EM density for the linker in the unsharpened maps (Fig. 4a), sharpening weakened this density to the extent where reliable placement of the main chain was not possible, indicating that the linker retains some flexibility.

**Interactions between Int and $C_t$ domains**. We found that the Int and $C_t$ domains form a relatively large contact interface (Fig. 5). The base of the Int domain (including the end of helix I, the beginning of the loop connecting helices I and II, and a large part of helix II) fits into the orifice of the substrate entrance tunnel of its partner $C_t$ and is simultaneously flanked by the tip of the oligomerization sub-domain of a second $C_t$ domain, which contacts the end of the loop between helices I and II and, to a lesser extent, helix III (Fig. 5a, b). This secondary interaction also occurs with a $C_t$ domain of the opposite dimer (e.g., Int of protomer A is docked into $C_t$ of protomer C and interacts with the oligomerization sub-domain of $C_t$ from protomer D). The total contact area between the Int domain (excluding the 4'-PP prosthetic group) and the two $C_t$ protomers is 653 Å$^2$. Calculation of electrostatic potentials revealed that the surface of the Int domain is mostly negatively charged, including the contact interface (Fig. 5c). This is in agreement with the known acidic nature of many A/PCPs[42]. In contrast, the corresponding contact area of the $C_t$ domain that accommodates 4'-PP is charged predominantly positively. The Int and $C_t$ domains form multiple

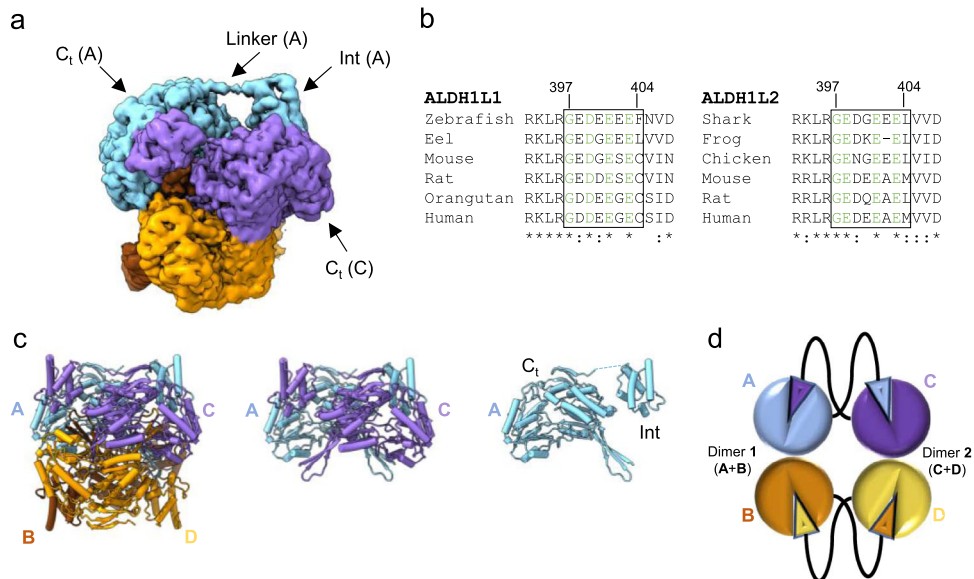

**Fig. 4 Pairing of intermediate (Int) and C-terminal ($C_t$) domains in ALDH1L1. a** Cryo-EM map of NADP$^+$-bound ALDH1L1 with the four protomers colored individually. Int of protomer A interacts with $C_t$ of protomer C. Within each protomer, Int and $C_t$ are connected with a long, extended linker. **b** Sequence alignments of the linker region for ALDH1L1 and ALDH1L2 family proteins. Invariant residues are colored *green*. **c** Pairing of Int and $C_t$ domains illustrated using *pipes-and-planks* representation of the atomic model. From left to right: the entire tetrameric structure (protomers A–D), two paired protomers (A and C), and a single protomer (A). **d** A schematic depicting pairing of the $C_t$ and Int domains of ALDH1L1. Each Int domain interacts with the $C_t$ domain of a protomer from a different dimer.

interactions, with the closest contacts between the main chain carbonyl of Gly351 (loop connecting helices I and II) and the amide of Gln693 (2.8 Å), the guanidinium group of Arg359 (helix II) and the side-chain oxygen of Asn745 (3.2 Å), as well as a side-chain oxygen of Glu366 (helix II) and the guanidinium group of Arg742 (3.3 Å). Alignment of the cryo-EM structures of ALDH1L1 and available X-ray structures of the $C_t$ domain revealed no significant alterations in positions of amino acid side chains in the regions that contact the Int domain. Similarly, structural superposition of the Int domain with the NMR structure of individual human Int without the 4′-PP prosthetic group (PDB 2cq8) revealed no major differences (r.m.s.d. = 0.83 Å). Therefore, complex formation between the Int and $C_t$ domains relies on the shape and charge complementarity.

**Interactions of the 4′-phosphopantetheine prosthetic group.** The 4′-PP group covalently linked to serine 354 of the Int domain was found in the fully extended conformation, penetrating deep into the substrate entrance tunnel of the $C_t$ domain and making multiple contacts with residues forming the tunnel (Fig. 6a). Two lysine residues of the $C_t$ domain, Lys520 and Lys865, formed ion pairs with the phosphate of 4′-PP. The main chain carbonyl oxygen of Asn864 and the hydroxyl of Thr521 were within the hydrogen-bonding distance from the hydroxyl and carbonyl oxygens, respectively, of the pantothenic acid moiety. The amide group of asparagine 706 was positioned 3.5 Å from the carbonyl of the β-alanine moiety of 4′-PP.

Curiously, we found that the sulfur atom of the catalytic nucleophile Cys707 in the ALDH active center was positioned closely to the sulfur atom of the 4′-PP, and the cryo-EM density between the two atoms appeared continuous (Fig. 6b). This suggested that a disulfide bond formed between the two atoms. To verify the presence of such a bond, we conducted trypsin digestion of ALDH1L1 followed by liquid chromatography-mass spectrometry (LC/MS). Analysis of the LC/MS data revealed readily detectable ions at $m/z = 1094.5$ ([M + 2H]$^{2+}$) and $m/z = 730.0$ ([M + 3H]$^{3+}$) that corresponded to a tryptic digestion product of

nominal mass 2187 Da, identical to the theoretical mass of 4′-PP-crosslinked peptides $^{350}$S-R$^{359}$ and $^{704}$G-R$^{712}$ (Fig. 6c). The subsequent collision-induced dissociations of these ions resulted in a pattern of MS peaks, the interpretation of which allowed unequivocal identification of the chemical structure of the parent ions (Fig. 6d). The most intense peaks in the MS$^2$ spectra resulted from the break of the labile phosphate moiety followed by the neutral loss of the phosphate group, either as phosphoric ($H_2PO_4$) or meta-phosphoric ($HPO_3$) acid (Δ mass 98 and 80 Da, respectively)[45]. The presence of a series of b- and y-ions unambiguously confirmed the amino acid sequences and sites of the 4′-PP covalent modification in the crosslinked peptides. While it was not possible to quantify the prevalence of the crosslinked peptide, its presence confirmed that the disulfide bond formed between the 4′-PP prosthetic group and Cys707 in a population of ALDH1L1 molecules. The inclusion of the disulfide bond in the molecular models improved the fit of the two sulfur atoms to the cryo-EM density.

To investigate the contribution of the 4′-PP arm to the interactions between the Int and $C_t$ domains, we designed a shorter version of ALDH1L1, containing only the Int and $C_t$ domains (termed int-$C_t$), and expressed it in *E. coli*. It was shown previously that ALDH1L1 produced in bacteria lacks the 4′-PP prosthetic group[29], and therefore only protein–protein contacts could contribute to the interactions between the Int and $C_t$ domains in int-$C_t$. NS-EM of int-$C_t$ resolved the $C_t$ core and, in some cases, Int domains adjacent to it (Fig. 5d). However, the sites of the $C_t$ core that had been invariably occupied by the Int domains in full-length ALDH1L1 (Fig. 2a) were predominantly vacant in the int-$C_t$ protein lacking 4′-PP. In agreement with such an arrangement, only the tetrameric $C_t$ core was resolved in the 2.7-Å crystal structure of int-$C_t$ expressed in bacteria, with no electron density present for the Int domains. This indicates that the interactions of the 4′-PP arm with residues of the substrate entrance tunnel are critical for the formation of a stable complex between the Int and $C_t$ domains of ALDH1L1.

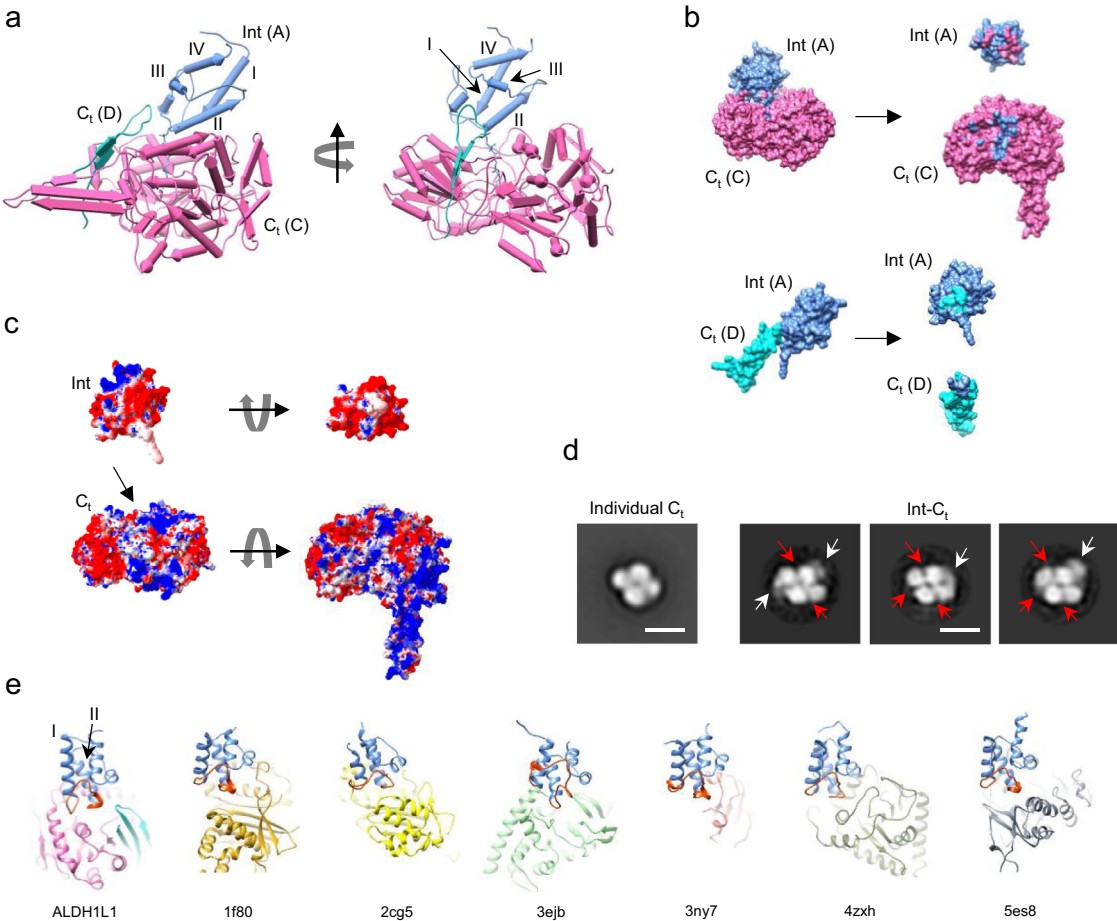

**Fig. 5 Interactions between the intermediate (Int) and C-terminal ($C_t$) domains of ALDH1L1. a** The intermediate domain (*light blue*) depicted in *pipes-and-planks* representation contacts $C_t$ domains of two protomers (*pink* and *cyan*). Only the oligomerization sub-domain is shown for the cyan protomer. The 4'-phosphopantetheine (4PP) arm is shown in *stick* representation. The alpha-helices of the Int domain are labeled I to IV. **b** Contacting surfaces of the Int domain and the two $C_t$ domains. Coloring is according to panel **a**. Left, surface representations of the complexes. Right, the complexes are split, and the domains are shown with the interfaces towards the viewer. Contact surfaces are colored according to the color of the interacting partner. **c** Surfaces of the Int and $C_t$ domains colored according to the electrostatic potentials, with *red* and *blue* corresponding to negative and positive charge, respectively. Left: the domains are shown in the same orientations as in the complex. The arrow points to the substrate entrance tunnel. Right: The same domains are rotated such that the contact interfaces face the viewer. **d** A comparison of negative-stain 2D classes of individual $C_t$ domain and the Int-$C_t$ protein expressed in bacteria. *White* arrows point to Int domains adjacent to the $C_t$ core. *Red* arrows point to vacant sites for binding of Int domain. Scale bars represent 10 nm. **e** A comparison of the complex between the Int and $C_t$ domains of ALDH1L1 with various complexes formed by acyl and peptidyl carrier proteins. The carrier proteins are aligned to the Int domain. The loop connecting helices I and II is colored *orange*. The PDB code is listed below each complex. 1f80: complex between Bacillus subtilis ACP and ACP synthase; 2cg5[85]: human fatty acid synthase ACP in complex with phosphopantetheinyl transferase; 3ejb[86]: *E. coli* ACP in complex with cytochrome P450; 3ny7[87]: *E. coli ACP* in complex with STAS domain of YchM anion transporter; 4zxh[84]: *Acinetobacter baumannii* PCP in complex with nonribosomal peptide synthetase; 5es8[88]: PCP in complex with adenylation domain of nonribosomal peptide synthase from *Brevibacillus parabrevis*.

**Int domains are highly mobile when not docked into $C_t$ domains**. During the initial step of the ALDH1L1 catalysis, 4'-PP arms must be accessible to interact with the $N_t$ domains. With the arm placed outside of the $C_t$ domain active site, the Int domain is expected to disengage the $C_t$ domain, leaving the substrate entrance tunnel vacant. However, global 3D classification did not produce classes with vacant substrate entrance tunnels of the $C_t$ domains. To determine the fraction of free (not occupying substrate entrance tunnels) Int domains, we performed local 3D classification within a mask encompassing each Int domain and calculated the total number of particles that contributed to empty versus occupied classes (Supplementary Fig. 2). This analysis produced estimated average occupancies of 76% and 83% for the Int domain in the ligand-free and NADP$^+$-bound structures, respectively. This indicates that a fraction of Int domains is present in the free form and is available to shuttle the substrate between

the $N_t$ and $C_t$ catalytic centers. However, since neither NS-EM nor cryo-EM experiments resolved Int in any other state than docked at the $C_t$ substrate entrance tunnel, the Int domains do not seem to assume strictly defined positions while shuttling between the two catalytic domains in the ligand-free and NADP$^+$-bound enzyme. These results also indicate that the state with the Int domains docked at the $C_t$ substrate entrance tunnels and with the 4'-PP arms reaching into the ALDH active sites is the most favorable conformation for the resting (substrate-free) enzyme. The partial disulfide bond between the 4'-PP arm and Cys707 could serve to support this conformation.

**Transient interactions of the N-terminal domain**. In an attempt to resolve the $N_t$ domain of ALDH1L1, we performed refinement of the cryo-EM dataset of ligand-free ALDH1L1 without

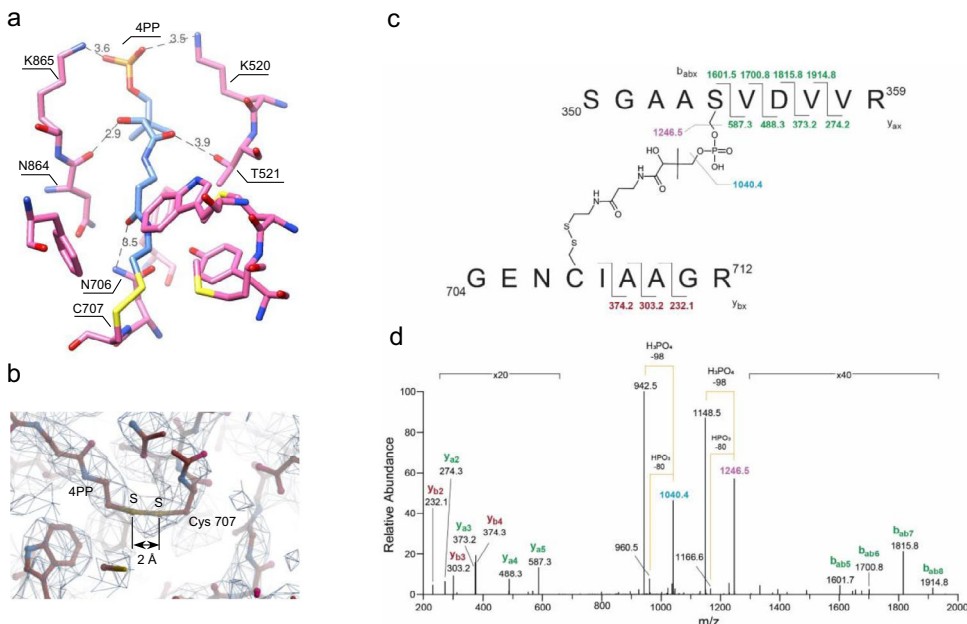

**Fig. 6 Contacts formed by the 4′-phosphopantetheine arm (4PP) of the intermediate domain of ALDH1L1. a** Interactions between 4PP (carbon atoms are colored *light blue*) and residues of the substrate entrance tunnel of the C-terminal domain (carbon atoms are colored *pink*). Inter-atomic distances shorter than 4 Å are labeled. **b** Cryo-EM density indicates the presence of a covalent bond between Cys707 and 4PP in the structure of apo-ALDH1L1. **c** Schematic representation of 4PP cross-linked peptides resulting from tryptic digestion of apo-ALDH1L1. **d** $MS^2$ spectrum of the MS ion at m/z 1094.5 $[M + 2H]^{2+}$. The fragmentation pattern reveals the presence of y- and b-ions characteristic of the amino acid sequence of the cross-linked peptides shown in panel c (breakdown along the peptide backbone). Importantly, the main ions observed in the spectrum correspond to the fragmentation along with the 4PP phosphate group, unveiling the chemical nature of the crosslink.

enforcing symmetry. The resulting 4.4-Å-resolution map was very similar to its symmetrical counterpart, except that weak density for a single $N_t$ domain became visible (Fig. 7a). This $N_t$ domain appeared to interact with the NADP$^+$-binding sub-domain of one of the $C_t$ domains. Low r.m.s.d. values between the chains of this structure (0.67–0.74 Å) indicated that this interaction did not induce large structural rearrangements in the protein core. A subsequent local 3D classification of the same dataset isolated a conformation with two clearly visible $N_t$ domains in the same orientation, corresponding to 5517 particles. 3D refinement of this smaller dataset resulted in a 6.8-Å map with strong density for the two $N_t$ domains. (Fig. 7b). The middle section of each $N_t$ domain was positioned within 8 Å from several secondary structure elements of the $C_t$ domain, and the amino-terminal portion of the $N_t$ domain sat directly above the α-helix (residues 653–664) of the $C_t$ domain that forms one side of the cleft accommodating the adenine moiety of NADP$^+$ [27]. Notably, although most α-helices of the $C_t$ core were well resolved in the map, there was no cryo-EM density for this key helix.

In another 3D class, containing 17,499 particles and refined to 7.0 Å resolution, a single $N_t$ domain was found to straddle the $C_t$ core between two Int domains, with both ends of $N_t$ in contact with the ALDH1L1 core (Fig. 7c). As expected, the carboxyl-terminal region of the $N_t$ domain was positioned close to the amino-terminal end of the Int domain of the same protomer. Interestingly, the amino-terminal portion of the $N_t$ domain interacted with the linker connecting the Int and $C_t$ domains of a different protomer. Although the resolution of the map is insufficient for interpreting this interaction at atomic details, it is clear that residues 55–61 of the $N_t$ domain, composing a loop and a short beta-strand, were supported by the Int-$C_t$ linker. Of note, while the linker is negatively charged (Fig. 3B), the complementary interface of $N_t$ contains a positively charged patch along the contact interface (Fig. 7c).

To confirm the formation of transient complexes between $N_t$ domains and the ALDH1L1 core, we performed chemical cross-linking of the full-length protein with 0.1% glutaraldehyde followed by NS-EM and 2D classification (Supplementary Fig. 3a). This treatment resulted in gradual disappearance of 2D classes displaying $N_t$ moieties not in contact with the protein core (Supplementary Fig. 3b), indicating that glutaraldehyde cross-linking stabilized the transient complexes formed by the N-terminal domains. Of note, $N_t$ domains attached to the core were reliably resolved in the 2D class averages even after prolonged glutaraldehyde treatment, suggesting that cross-linking occurred at specific positions.

## Discussion

ALDH1L1 has two catalytic centers located in separate domains and utilizes a carrier protein (evolutionarily incorporated as a domain) to transfer the reaction intermediate between these centers. This carrier protein domain is highly similar to A/PCPs employed in the biosynthesis of fatty acids, non-ribosomal peptides and polyketides, reactions performed by large and complex multi-enzymatic molecular machines[46–48]. This type of modular organization implies extensive domain movements accompanying the transport of substrate between the active sites. Likewise, we found that the tetrameric aldehyde dehydrogenase module of ALDH1L1, located at the C terminus, forms the rigid core of the enzyme, whereas the N-terminal hydrolase domains assume a continuum of positions apparently constrained mainly by the length of the inter-domain linkers. In our cryo-EM structures of ALDH1L1 in the resting state (i.e., in the absence of substrate), the Int (carrier) domains were resolved docked at the substrate entrance tunnels of the $C_t$ core, but the incomplete occupancy of these anchored carriers indicates that they operate as highly mobile units. The complex between the $N_t$ and Int domains was

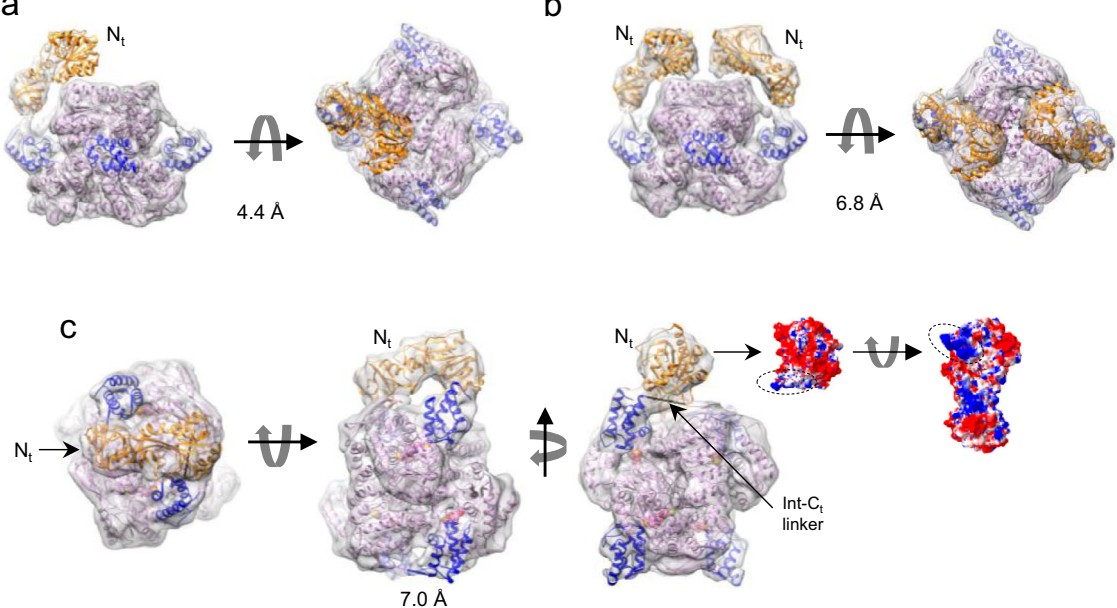

**Fig. 7 Transient interactions of the N-terminal ($N_t$) domain of ALDH1L1. a** Relaxation of the D2 symmetry led to the appearance of weak density for one $N_t$ domain in the cryo-EM map of ligand-free ALDH1L1. **b** Conformation with two symmetrically located $N_t$ domains. The $N_t$ domains interact with the regions of the $C_t$ core responsible for the binding of NADP$^+$. **c** Conformation of ALDH1L1 with one $N_t$ domain interacting with the linker between the intermediate and the $C_t$ domains (Int-$C_t$ linker) of a different protomer. The linker is represented with the *black spring*. The 4′-PP arms are in *sphere* representation. In the two images to the right, the surface of the $N_t$ domain is colored according to the electrostatic potential. The positively charged patch interacting with the Int-$C_t$ linker is enclosed in *dashed* ovals. In the remaining images, the $N_t$, Int, and $C_t$ domains are colored *gold*, *blue*, and *salmon*, respectively.

not detected by cryo-EM or NS-EM, which suggests it is transient in nature, a common phenomenon for interactions involving A/PCPs[49]. Overall, our findings provide strong structural support for the catalytic mechanism wherein (i) the $N_t$ domain transfers the formyl group from 10-fTHF to the 4′-PP arm of the Int domain and (ii) the Int domain delivers the formyl to the $C_t$ domain where (iii) the formyl is oxidized to $CO_2$[29] (Supplementary Fig. 4). Of note, in vitro both the $N_t$ and $C_t$ domains, either expressed individually or within the full-length enzyme, are capable of independent catalysis, 10-formylTHF hydrolysis or small chain aldehyde oxidation, respectively[26,34]. It is not clear whether such independent activities take place in the cell since the hydrolase catalysis in vitro requires high concentrations of non-physiological sulfhydryls while putative substrates for the ALDH reaction are unknown. Thus, the complex mechanism enabled by the merging of the three domains is likely the only catalytic function of ALDH1L1.

Most aldehyde dehydrogenases are known to exist as either homodimers, typified by the members of the ALDH3 family, or homotetramers, represented mainly by ALDH1/2 families, which also include ALDH1L1[50]. Such homotetramer is organized as a dimer of homodimers formed by protomers A/B and C/D as schematically presented in Fig. 4d. Although the enzymatic mechanism of ALDH1L1 does not dictate a specific quaternary organization of the enzyme, the 3-dimensional structure revealed that the tetrameric state of the C-terminal ALDH module is indispensable for the enzyme function. In our cryo-EM structures, the carrier domains of protomers A and B were paired with the ALDH domains of protomers C and D, respectively, while the carrier domains of protomers C and D interacted with the $C_t$ domains of protomers A and B. This pairing scheme can only be realized in a tetrameric enzyme. Furthermore, we found that the length and composition of the linker connecting the Int and $C_t$

domains are preserved in cytosolic (ALDH1L1) and mitochondrial (ALDH1L2) enzymes as well as across multiple species, suggesting that this intricate domain pairing is a universal characteristic of 10-formyltetrahydrofolate dehydrogenases. Of note, although multiple studies analyzed the oligomeric state of ALDH proteins (recently reviewed in Shortall et al.[51]), the physiological significance of oligomerization is unclear for most of these enzymes. One possible exception is tetrameric ALDH from *Thermus thermophilus*, which has a ~30 amino-acid-long C-terminal extension that interacts with the N-terminal region of a protomer from a different homodimer[52]. Other ALDHs, including fatty aldehyde dehydrogenase (FALDH) and ALDH7A1, also feature short C-terminal extensions, but they interact with the protomer within the same homodimer[53,54]. In contrast, ALDH1L1 is the example of an ALDH with two additional domains spanning 400 amino acids at the N-terminus of the enzyme, with the tetrameric state being a prerequisite for its complex function.

The acyl and peptidyl carrier proteins evolved to shuttle catalytic intermediates between reaction centers, which requires interaction with multiple partner proteins. This functional versatility necessitates that the nature of such interactions is transient, which is crucial for the uninterrupted action of molecular machines that employ A/PCPs[49]. Accordingly, the contact area between the A/PCP and the partner protein is usually small, with most interacting residues confined to helix II of the carrier protein and, to a lesser extent, helix III and the part of the linker between helices I and II that is close to helix II[42,55–57]. The small size of the contact interface often requires cross-linking to enable structural investigation[57–59]. In contrast, the relatively large contact interface between the Int and $C_t$ domains of ALDH1L1 was resolved in its native, non-cross-linked form (Fig. 5a–c). In addition to helix II, this interface also involves the base of helix I.

Moreover, distinct to other A/PCPs, the beginning of the loop connecting helices I and II protrudes towards the $C_t$ domain to interact with two helices forming the orifice of the substrate entrance tunnel (Fig. 5e). Thus, the structure of ALDH1L1 illustrates, to our knowledge, a new mode of interaction between an A/PCP-like carrier protein and its catalytic partner. Importantly, while this interaction favors the state with Int docked at the ALDH substrate entrance tunnel, the incomplete occupancy of this anchored Int domain indicates that this interaction is reversible and does not preclude the shuttling of free (undocked) Int between the catalytic domains during catalysis.

Importantly, the 4′-PP arm of the carrier domain was fully resolved in our cryo-EM maps. It spanned the entire 12-Å-deep substrate entrance tunnel of the ALDH domain, extending towards the catalytic cysteine. We found that the contacts formed by the 4′-PP prosthetic group are critical for the formation of a stable complex between the Int and ALDH domains of ALDH1L1. During catalysis, the extended 4′-PP conformation would place the formyl group transported from the $N_t$ domain precisely in the ALDH active site, allowing the nucleophilic attack by Cys707. Curiously, in the absence of the substrate, a partial covalent bond formed between the sulfur atoms of 4′-PP and Cys707. While formation of this disulfide link is likely prevented by the formyl group attached to the 4′-PP arm during catalysis, it could be hypothesized that in the resting state of the enzyme such a bond prevents irreversible oxidation of both the catalytic cysteine and 4′-PP sulfur atoms. This disulfide could be reduced by cellular glutathione accessing the active center through the $NADP^+$ binding site. Of note, in the individually expressed $C_t$ domain, Cys707 was shown to form a transient covalent adduct with the C4 atom of the nicotinamide ring of $NADP^+$ [27], which suggests that this cysteine is highly reactive beyond the immediate catalytic step. It could also be hypothesized that maintaining the 4′-PP arm within the substrate entrance tunnel prevents the entrance of small aldehydes into the ALDH catalytic center in vivo, thus preserving the enzyme for the 10-formylTHF dehydrogenase catalysis. Alternatively, we cannot exclude the possibility that the observed disulfide bond is the result of oxidation in our experimental setting.

The high mobility of the $N_t$ domains suggests that ALDH1L1 catalysis is driven primarily by stochastic domain movements. However, the cryo-EM maps of the states with $N_t$ domains resolved in fixed positions provide evidence of non-random interactions of these functional modules, which could play a role in the enzymatic mechanism. In one such cryo-EM map, two $N_t$ domains were shown to interact with the $NADP^+$ binding regions of the ALDH domains (Fig. 7b). Since a key helix forming the $NADP^+$ binding site was disordered in each involved ALDH domain, it is possible that in this conformation the $N_t$ domain interferes with the binding of $NADP^+$. In support of such a possibility, this conformation was not detected in the ALDH1L1-$NADP^+$ dataset. Based on these results, we hypothesize that the hydrolase domains of ALDH1L1 may be involved in regulating the enzymatic reaction performed by the ALDH domains. It has to be noted, however, that full-length ALDH1L1 and the individually expressed ALDH domain displayed similar affinities for $NADP^+$ ($K_d$ of 0.3 μM versus 0.2 μM, respectively)[26], suggesting that the proposed effect is likely small. In the second cryo-EM map, the N-terminal moiety of the hydrolase domain rested on the linker connecting the carrier and ALDH domains of a different protomer (Fig. 7c), with a remarkable charge complementarity between the linker, carrying a strong negative charge (Fig. 4b), and the positively charged region of the $N_t$ domain contacting it (Fig. 7c). We surmise that in this ALDH1L1 conformation the $N_t$ domain may be involved in the extraction of the Int domain from the substrate entrance tunnel of the $C_t$ core, with

the positively charged patch acting as a hook. Alternatively, this domain arrangement may create a scaffold for the formation of the complex between the $N_t$ and Int domains. In the latter scenario, a large-scale rotation and shift of the Int domain would be necessary to bring together the sulfhydryl of the 4′-PP arm and the $N_t$ active site residues, which are ~50 Å apart. While the above interpretations are speculative, the existence of scarcely populated states with firmly positioned hydrolase domains alludes to an intricate mechanism of catalysis that may involve various auxiliary inter-domain interactions guiding the overall random domain movements during catalysis.

In summary, in this study cryo-EM revealed the unusual architecture of the multi-domain enzyme ALDH1L1, which enables the complex catalytic mechanism. Protein oligomerization and multidomain organization are common phenomena in eukaryotes[60–63]. While a modular organization can expand the enzyme functionality[61], oligomerization provides benefits such as efficiency, regulation and stability[63]. In some cases, oligomerization is required because catalytic centers are formed by residues from different protomers or because oligomers enable additional non-catalytic regulatory sites. Metabolic enzymes can also form structures of higher degree of order like filaments, which might not directly affect the catalysis within a single unit[64]. Of note, all such examples were reported in folate metabolism where ALDH1L1 belongs[65–68]. Here we uncovered another mechanism in which tetrameric organization allows modular catalysis bypassing spatial restrictions within a single protomer. Thus, the tetrameric state of ALDH1L1 is indispensable for the enzyme functionality, which also involves transient domain interactions and large-scale domain movements. Finally, the complex between the intermediate and aldehyde dehydrogenase domains of ALDH1L1 demonstrates, to our knowledge, a new mode of interaction between an A/PCP-like carrier protein and a catalytic domain, emphasizing the versatility of A/PCPs.

## Methods

**Protein expression and purification**. Full-length rat ALDH1L1 was expressed following a previously developed protocol[69]. Specifically, High Five insect cells (Invitrogen) grown as monolayer (Grace's insect medium supplemented with 10% fetal bovine serum/175-cm² cell culture flasks) at 27 °C were infected with a high titer recombinant baculovirus stock produced as previously described[69]. Five days after infection, the culture medium was collected, and detached cells were removed by centrifugation (10,000 × g, 10 min). To purify ALDH1L1, the cell culture medium was applied to a column containing 5-formyl-THF-Sepharose affinity resin equilibrated with 10 mM Tris-HCl buffer, pH 7.4, containing 10 mM 2-ME and 1 mM NaN₃ (buffer A). The column was washed with buffer A and then with the same buffer containing 1.0 M KCl; the enzyme was eluted with buffer A containing 1.0 M KCl and 20 mM folic acid. The eluate was concentrated and excess KCl removed using a spin concentrator. Additional purification was then carried out using FPLC/Mono-Q column (GE) chromatography with a linear KCl gradient (0–0.5 M in buffer A) and Sephacryl S-300 (GE) size-exclusion chromatography in buffer A with 0.2 M NaCl. The individual $C_t$ domain and Int-$C_t$ protein were expressed as 6xHis tagged constructs in E. coli (Invitrogen) from pRSET vectors. Protein expression was carried out at 22 °C, and the soluble cell fraction was separated by sonication and centrifugation. The proteins were purified using Ni-NTA or Co-NTA agarose (Qiagen) using a 5–20 mM imidazole gradient to remove impurities followed by elution with 100 mM imidazole in buffer A supplemented with 100 mM KCl. Additional purification was done by size-exclusion chromatography on Sephacryl S-300. The purity of all proteins was confirmed by SDS-PAGE with Coomassie staining. Purified full-length ALDH1L1 was tested for the 10-formylTHF dehydrogenase activity as we previously described[33]. $C_t$ domain and Int-$C_t$ protein were tested for the aldehyde dehydrogenase activity using propanal as the substrate and $NADP^+$ as the cofactor essentially as we described[26]. After purification, all protein preparations used in the present study had specific activities close to previously reported values[26,33] and were stored at −80 °C in the presence of 10 mM 2-ME and 20% glycerol.

**Liquid chromatography/mass spectrometry**. In total, 30 μg of ALDH1L1 (50 μL of protein solution) was combined with 25 μL of 9 M urea and 10 μL of acetonitrile and incubated for 10 min at 42 °C. This mixture was diluted with 250 μL of 100 mM ammonium bicarbonate prior to the addition of 5 μg of sequencing grade trypsin (Promega). The proteolytic digestion was carried out for 4 h at 37 °C. The

resulting peptides were loaded onto a reverse-phase C4 (2.1 mm × 50 mm) column (Thermo Scientific). Peptides were resolved and eluted with a gradient of acetonitrile in water (from 98% $H_2O$ with 0.1% (v/v) formic acid (A) and 2% acetonitrile with 0.1% (v/v) formic acid (B) to 100% B) developed over 20 min. Separation was achieved at a flow rate of 0.3 mL/min using an Agilent Technology 1100 Series HPLC system. The eluent was directed into an LTQ Velos linear trap quadropole mass spectrometer (Thermo Scientific) equipped with an electrospray ionization source operated in positive ion mode. Parameter settings of the mass spectrometer for peptide detection were as follows: activation type, collision-induced dissociation; normalized collision energy, 35 kV; capillary temperature, 370 °C; source voltage, 5 kV; capillary voltage, 43 V; tube lens, 105 V. MS spectra were collected over a 200–2000 m/z range. The raw MS data were analyzed using Qual Browser for Thermo Xcalibur version 2.1.

**Negative-stain electron microscopy.** Protein samples were diluted with buffer containing 10 mM HEPES, pH 7, and 150 mM NaCl to ~0.02 mg/ml. A 4.7-μl drop of the diluted sample was placed on a freshly glow-discharged carbon-coated copper grid and left for 15 s. Excess liquid was removed using filter paper, and the grid was washed three times with 4.7-μl drops of the same buffer. After the final wash, the buffer drop was removed in the same manner, and the protein was negatively stained by applying a 4.7-μl drop of 0.75% uranyl acetate for 30 s. Excess negative stain was removed using filter paper, and the grid was allowed to dry. Data were collected using SerialEM[70] on a Tecnai T20 electron microscope (FEI, the Netherlands) equipped with a $LaB_6$ filament operated at 200 kV and a 2k × 2k FEI Eagle CCD camera. The nominal magnification was 100,000x, which corresponded to a pixel size of 2.2 Å. EMAN2[71] was used to semi-automatically select 249,416 particles from 3920 micrographs. The selected particles were extracted into 128 × 128-pixel boxes and subjected to reference-free 2D classification into 256 classes using Relion 2.1[72]. For separate visualization of the $N_t$ domains, 30,186 peripheral domains (arms) of negatively stained FDH molecules were selected manually using EMAN2 from 392 micrographs of the same dataset. The selected particles were extracted into 64×64-pixel boxes and classified using Relion 2.1 into 256 classes.

**Chemical cross-linking and comparative quantification of mobile N-terminal domains.** Full-length ALDH1L1 was diluted to 0.01 mg/ml with buffer containing 10 mM HEPES, pH 7, and 150 mM NaCl, followed by the addition of 0.1% glutaraldehyde. Aliquots were taken before the addition of glutaraldehyde and after 1 min, 5 min, 10 min, 30 min, and 60 min of incubation at 4 °C, and negative staining and NS-EM data collection were performed as described above. All datasets were subjected to 2D classification in Relion. After discarding 2D classes that did not represent intact ALDH1L1 molecules, the final NS-EM datasets contained 57,888 (control), 36,538 (1 min), 32,722 (5 min), 39,150 (10 min), 47,877 (30 min), and 30,449 (60 min) particles. 2D class averages displaying ALDH1L1 molecules with at least one arm that was not in contact with the protein core were identified by visual inspection, and their fractions were calculated based on the total number of particles that contributed to these classes. These fractions were used solely for the purpose of comparing the cross-linking time points because not all mobile $N_t$ domains could be captured by 2D classification due to their dynamic nature.

**Cryo-electron microscopy specimen preparation and data collection.** ALDH1L1 was vitrified at a concentration of 0.4 mg/ml in 20 mM HEPES, pH 7.6 (apo-ALDH1L1) or 40 mM HEPES, pH 7, 1 mM $NADP^+$ (ALDH1L1-$NADP^+$ complex). Cryo-EM specimens were prepared by plunge-freezing in liquid ethane using Vitrobot Mark IV (FEI) at room temperature and 90% humidity. The grids (Quantifoil R2/2 with gold support) were glow-discharged for 30 s at a pressure of 37 mBar and with the current set to 30 mA. The drop volume was 3 μl. Data were collected at the National Cryo-Electron Microscopy Facility (NCEF) at National Cancer Institute on a Titan Krios electron microscope (FEI) operated at 300 kV and equipped with a K2-Summit direct electron detector (Gatan). The detector was used in the super-resolution mode. For apo-ALDH1L1, 2202 movies were collected with a nominal dose of 40 $e^-/Å^2$ equally distributed between 40 frames of a 12-s movie, and the pixel size (super-resolution mode) was 0.66 Å (magnification: 105,000x). The defocus range was −1 to −3 μm. For ALDH1L1-$NADP^+$ complex, 2381 movies were collected with a nominal dose of 40 $e^-/Å^2$ equally distributed between 40 frames of a 14-s movie, and the pixel size (super-resolution mode) was 0.532 Å (magnification: 130,000x). The defocus range was −1 to −2.5 μm (Table 1).

**Single-particle analysis of cryo-electron microscopy data.** Motion correction and dose weighting were performed using MotionCor2[73]. For local motion correction, frames were divided into 25 tiles. Images were binned 2x (apo-ALDH1L1) and 1.5x (ALDH1L1-$NADP^+$) during motion correction, resulting in pixel sizes of 1.32 Å and 0.76 Å, respectively. Contrast transfer function parameters were estimated using ctffind 4.1[74]. All other image processing steps were performed in Relion 3.0[72] unless stated otherwise. Particles were picked automatically using projections of an X-ray structure of the tetrameric C-terminal domain of ALDH1L1 (PDB 2o2p[27]) low-pass filtered to 40 Å, resulting in datasets containing

**Table 1 Cryo-EM data collection and single-particle analysis statistics.**

| | Ligand-free ALDH1L1 EMDB-24540 PDB 7RLT | ALDH1L1-NADP$^+$ complex EMDB-24547 PDB 7RLU |
|---|---|---|
| **Data collection and processing** | | |
| Magnification | 105,000 | 130,000 |
| Voltage (kV) | 300 | 300 |
| Electron exposure (e$^-$/Å$^2$) | 40 | 40 |
| Defocus range (μm) | −1.0 to −3.0 | −1.0 to −2.5 |
| Pixel size (Å) | 0.66 | 0.53 |
| Symmetry imposed | D2 | D2 |
| Initial particle images (no.) | 1,082,600 | 1,050,740 |
| Final particle images (no.) | 86,276 | 202,398 |
| Map resolution (Å) | 3.7 | 2.9 |
| FSC threshold | 0.143 | 0.143 |
| Map resolution range (Å) | 2.9–6.0 | 2.0–4.0 |
| **Refinement** | | |
| Initial model used (PDB code) | 2o2p | 2o2p |
| Model resolution (Å) | 3.8 | 3.0 |
| FSC threshold | 0.5 | 0.5 |
| Map sharpening B factor (Å$^2$) | −186.4 | −113.6 |
| Model composition | | |
| Non-hydrogen atoms | 17,996 | 18,188 |
| Protein residues | 2,324 | 2,344 |
| Ligands | 4 | 8 |
| Water | 0 | 28 |
| B factors (Å$^2$)(mean) | | |
| Protein | 91.4 | 16.2 |
| Ligand | 79.5 | 20.7 |
| Water | N/A | 10.9 |
| R.m.s. deviations | | |
| Bond lengths (Å) | 0.005 | 0.005 |
| Bond angles (°) | 0.877 | 0.901 |
| Validation | | |
| MolProbity score | 1.50 | 1.00 |
| Clash score | 0.86 | 0.25 |
| Poor rotamers (%) | 1.61 | 0.31 |
| Ramachandran plot | | |
| Favored (%) | 88.73 | 94.00 |
| Allowed (%) | 10.27 | 5.31 |
| Disallowed (%) | 1.00 | 0.69 |

*FSC* fourier shell correlation, *PDB* Protein Data Bank, *R.m.s.* root mean square.

1,082,600 (apo-ALDH1L1) and 1,050,740 (ALDH1L1/$NADP^+$) particles. The particles were extracted, with 2x binning, into 80 × 80 (apo- ALDH1L1) or 140 × 140 (ALDH1L1/$NADP^+$) pixel boxes and subjected to reference-free 2D classification into 128 classes with selection of high-resolution classes corresponding to a complete, undistorted tetramer of the $C_t$ domain that appeared symmetrical. This selection reduced the size of the datasets to 424,239 and 640,945 particles, respectively. The corresponding particles were re-extracted, without binning, into 160 × 160 (apo-ALDH1L1) or 280 × 280 (ALDH1L1/$NADP^+$) pixel boxes, and reference-free 2D classification into 128 classes was repeated. Selection of best-looking classes resulted in datasets of 147,837 (apo-FDH) and 594,883 (FDH/$NADP^+$) particles. 3D classification into 10 classes was performed next with the above-mentioned X-ray structure of the tetramer of the $C_t$ domain low-pass filtered to 40 Å serving as the initial model. No symmetry was imposed at this stage. The presence of additional density at the substrate entrance tunnel of the $C_t$ domain was obvious in the resulting 3D classes, and in all 3D classes with sufficiently high resolution this density consisted of four α-helices and an arm protruding deep into the substrate entrance tunnel. Additional density consistent with the size and shape of the $N_t$ domain of ALDH1L1 (PDB 1s3i[21]) was observed in several 3D classes. High-resolution 3D classes, as well as 3D classes with density for one or more $N_t$ domains, were subjected to 3D auto-refinement with D2 or C2 symmetry imposed as well as without imposing symmetry. Post-processing included automatic B-factor sharpening and detector modulation transfer function correction, and the gold-standard resolution was determined within a soft mask using a 0.143 FSC threshold. Local resolution was estimated using ResMap[75]. Representative micrographs and 2D class averages, FSC curves, and local resolution

data are presented in Supplementary Figs. 5 and 6. Supplementary Fig. 7 illustrates the cryo-EM density for 4′-PP and NADP⁺.

**Model building**. The crystal structure of the tetrameric C-terminal domain of ALDH1L1 (residues 405-902) in the apo form (PDB 2o2p[27]) or in complex with NADP⁺ (PDB 2o2q[27]) and four instances of a homology model of the Int domain of rat FDH (residues 306–402) obtained using the SWISS-MODEL server[76] were fit into the corresponding cryo-EM density using USCF Chimera[77]. This was followed by one round of real-space refinement in PHENIX[78] and alternating rounds of model building in Coot[79] and restrained model refinement in Refmac[80]. Molprobity[81] was used to assess the quality of the atomic models. Map-model correlations were evaluated using phenix.mtriage[82].

**Estimation of occupancy of Int domains**. A soft mask was prepared for each of the four Int domains by segmenting the symmetrical ALDH1L1 map in UCSF Chimera (Supplementary Figure 2). Before 3D classification, both apo-ALDH1L1 and ALDH1L1-NADP⁺ maps were refined without symmetry imposed. 3D classification into 8–10 classes without particle alignment was then performed in Relion 3.0 for each Int domain separately using the final map low-pass filtered to 40 Å as the reference. The resulting 3D classes were examined visually, and total particle counts for classes with occupied and vacant Int domain binding sites were determined. Int domain occupancy was calculated as the fraction of the particles contributing to the classes representing occupied sites, averaged across the four sites within the tetramer.

**Other methods**. Protein structure similarity search was performed with the mTM-align server (35). Figures were prepared in UCSF Chimera, UCSF ChimeraX[83], and Coot.

**Statistics and reproducibility**. LC/MS experiments were repeated four times. The cross-linked peptides were detected in all these experiments.

**Reporting summary**. Further information on research design is available in the Nature Research Reporting Summary linked to this article.

## Data availability

Cryo-EM maps of ligand-free ALDH1L1 and ALDH1L1 in complex with NADP⁺ have been deposited to the EMDB with accession codes EMDB-24540 and EMDB-24547, respectively. Fitted coordinates have been deposited to the PDB with accession codes 7RLT and 7RLU, respectively. All other data are available from the corresponding authors upon request.

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

## Acknowledgements

This work was supported in part by federal funds from the Frederick National Laboratory for Cancer Research, NIH, under Contract HHSN261200800001 (YT) and the NIH R01 DK54388 grant (SAK). M.G. was supported by the NIH R01 EY023948 grant. Cryo-EM datasets were collected at the National CryoEM Facility (NCEF) of the National Cancer Institute. We would like to thank Dr. Ulrich Baxa for collecting cryo-EM data. This research was, in part, supported by the National Cancer Institute's

National Cryo-EM Facility at the Frederick National Laboratory for Cancer Research under contract HSSN261200800001E.

## Author contributions

Y.T. and S.A.K. conceived and planned the research. V.S. and N.I.K. produced the proteins. Y.T. and V.S. performed negative-stain EM and prepared cryo-EM specimens. Y.T. performed single-particle analysis of negative-stain and cryo-EM datasets. M.G. performed LC/MS experiments. Y.T. and S.A.K. wrote the manuscript, with all authors providing revisions and comments.

## Funding

## Competing interests

The authors declare no competing interests.
