## [Transparent Peer Review File · Communications Biology]

Reviewers' comments:

Reviewer #1 (Remarks to the Author):

The original article entitled "Structural insight into the function of putative tumor suppressor ALDH1L1" for publication in Nature Communications Biology describes an in-depth analysis of the structural aspects of the ALDH1L1 protein comprised of 3 distinctive domains that work together to perform catalysis. The manuscript delves into the interactions of the domains and the possible functional roles that these interactions play. This study allows insight into the versatility of ALDH structure, essential for function. The enzyme is equipped with two catalytic domains at the N-terminus (folate binding and hydrolase) and C-terminus (traditional ALDH enzyme) linked by an essential carrier protein domain that shuttles the reaction intermediates from N to C-terminal active site. While the three domains' structures have been resolved individually, this is the first report of the overall structure of the ALDH1L1 in totality, allowing understanding of key structural interactions. This paper is of great interest to ALDH enthusiasts, enzymologists and structural biologists alike.

Abstract and Introduction

The abstract and introduction provide good insight to the manuscript.

Results and Discussion

Results are well presented, some suggestions follow for further discussion.

- 1) The source of the enzyme ALDH1L1 that is being studied is not mentioned in the main body of the manuscript and should be included in the first paragraph of results.
- 2) Further discussion and speculation of certain aspects would add to the paper. Is there any previous reports on the substrate specificity of ALDH1L1, can it carry out ALDH dehydrogenase activity independent of its other domains? Does the Int domain play a role in stopping spontaneous ALDH activity independent of the other two domains?
- 3) The paper lacks comparison to other ALDH members. Discussion of other ALDHs that form tetramers in the same cross-over manner as the Int and Ct domains, e.g. ALDH from *Thermus thermophilus*.
- 4) Other ALDHs that use structural extensions for active site regulation should be discussed, e.g. ALDH7A1, FALDH with a gatekeeper (both from human) and ALDH from *T. thermophilus*.
- 5) It should be proposed whether the Int domain docking in the active site of ALDH potentially plays a role in active site regulation for entry of substrate in ALDH1L1.
- 6) Comment on the role of oligomerization of the Int and Nt domains in more detail.
- 7) Has the expressed and purified full length protein been trialled for enzymatic activity? Can the conversion of substrate to product be obtained with a mixture of individual domains in solution or is the combination of domains and tetrameric assembly essential.
- 8) The reviewer suggests making a more distinct colour contrast being domains for figure 2d, 2e and 7 to make visualisation clearer. For figure 3c, the subdomains of the ALDH domain could be coloured differently rather than just labelled.

Other minor corrections:

Line 192-198: Should this structural superimposition refer to a figure

Line 390: 30 µg not thirty µg

Reviewer #2 (Remarks to the Author):

In this paper, Tsybovsky et al. report a structural analysis of the rat ALDH1L1. This is a well written manuscript and the results are novel and interesting. The structure of the complete protein obtained by Cryo-EM revealed important mechanistic features of this enzyme, notably the conformational flexibility of the hydrolase domain and the interaction between the phosphantetheine group of the

A/PCP domain and the active site of the ALDH domain. I believe that this is an excellent piece of work, which significantly advances the knowledge of this complex enzyme and merits to be published. However, in my opinion, there are a few weak points that the authors need to consider before publication:

1) They found a disulfide bond between the catalytic Cys of the ALDH domain and the sulfur of the 4'-PP prosthetic group in a population of the enzyme molecules and hypothesize that formation of this covalent bond may be a mechanism of protection against oxidation of the highly reactive thiols of the catalytic cysteine and 4'-PP in the absence of the substrate. However, the formation of this disulfide would irreversibly inactivate the ALDH and sequester 4'-PP, so it does not provide any protection. Different from a thiol-nicotinamide adduct, which is reversible, a disulfide bond can only be broken by disulfide thiol exchange, and it is difficult to envision that physiological thiols, such as reduced glutathione for instance, could reach the position of this disulfide deep into the ALDH active-site tunnel. In my opinion, this disulfide is indeed the result of the oxidation of the two thiols during the experiments. In the text there is no mention to the inclusion of a reducing agent in the buffers used during the purification of the protein, its storage or in the protein solution used in the Cry-EM experiments. Perhaps if such compounds were included in the protein solution, and/or if the Cry-EM experiments were carried under anaerobic conditions if possible, the disulfide would have not been formed.

2) Fig. 1: The formyl group in the 10-formyltetrahydrofolate formula should have a hydrogen attached to the carbonyl C=O; delete the line, or substitute it for H, so that the aldehyde group is clearly seen. Formic acid is not formed as an intermediate of the reaction. Show the correct formaldehyde intermediate and the 4'-PP bound to this intermediate.

Minor points:

- 1) Introduction, line 79: 4'-PP is bound through a phosphoester bond, not by a phosphodiester as said.
- 2) Results, line 191: 'the amino group of Arg359' should be 'the guanidine (or guanidinium) group of Arg359'.
- 3) Give a more detailed description of the protein purification method; the work cited as this respect does not give such details.

Reviewer #3

The manuscript by Tsybovsky et al describes the full-length structure of the 10-formyl-THF dehydrogenase ALDH1L1 by high-resolution cryo-EM, building on previous structures of the individual domains of ALDH1L1. The structure is clearly of biological significance considering that ALDH1L1 is a key metabolic enzyme and tumor suppressor, and shows an intriguing oligomeric architecture with complex interactions among ALDH1L1 domains. Furthermore, the structure is of value to a broader community as it serves as a model for multi-domain enzymes with multiple reaction centers, as well as a model of complex oligomeric enzyme architecture.

The major claims of the paper are:

- 1) The intermediate (Int) and C-terminal ALDH (Ct) domains form the stable tetrameric core of the enzyme, while the N-terminal hydrolase (Nt) domains form highly mobile structures at the periphery of the structure.
- 2) The intermediate (Int) and C-terminal (Ct) domains from different protomers interact via broad electrostatic surfaces in a criss-crossing arm fashion to stabilize the Int-Ct tetrameric core. The interface is generally similar to other A/PCP proteins and their interaction partners, but is unusual in terms of its broad surface area.
- 3) The 4'-phosphopantetheine (4'-PP) prosthetic group of the Int domain further stabilizes Int-Ct domain interaction by inserting into the substrate entrance tunnel of the Ct-domain, and extending toward the catalytic cysteine.

- 4) The insertion of 4'-PP into the Ct domain substrate entrance tunnel suggests a mechanism whereby the 4'-PP group delivers the formyl group from the Nt domain to the catalytic cysteine of the Ct domain in an active state and prevents the oxidation of the catalytic cysteine in the resting state.
- 5) Despite the flexibility of the Nt domain with respect to the Int-Ct core, the N-terminal domain forms transient interactions with the Int-Ct body, specifically the Int-Ct linker and the NADP+ binding surface of the Ct domain.

Overall, the structure is elegant and the manuscript provides an articulate description of the novel structure. Most of the claims, especially 1)-3) are convincing given the relatively high resolutions of the structures. However, the main weakness of the work is a lack of any other data besides the cryo-EM structure to support the authors' interpretations. There are no functional assays or mutational structure-function studies that could backup the authors' arguments that the flexibility of structural element X or the domain interaction of Y&Z are important for enzyme function. The potential interactions between the different domains are intriguing, but the functional significance of those interactions is unclear without supporting experiments. If such additional data cannot be provided, at the very least, some of the claims on the importance of certain structural elements such as the Int-Ct linker, or the docking of the Nt domain and the tetrameric core should be toned down as more tentative ideas. In addition, the manuscript might be further improved if the authors can perform additional structural analyses of the cryo-EM data as explained below.

Some of my major suggestions are summarized as follows:

1. The authors suggest that the cryo-EM structure reflects the "resting state" of the enzyme. This is plausible considering that the Int domain & 4'-PP is plugging the substrate entrance tunnel all the way to the catalytic Cys without any substrate. However, the burning question in my mind is whether the full-length enzyme used in the structural studies has any enzymatic (hydrolase/dehydrogenase) activity, i.e., is the ALDH1L1 used in the study competent to assume an active state upon addition of substrate?

If so, is there any gross structural change in ALDH1L1 upon substrate addition?

If not, why is the enzyme inactive, and what other factors are required to transition ALDH1L1 to an active state?

2. The authors impose D2 symmetry on the cryo-EM data processing to get to a higher-resolution structure, which itself is a fully valid method, but I wonder whether valuable information is being lost by imposing symmetry. When relaxing or omitting the symmetry constraints, do the authors find any differences among subunits in the Ct & Int domains? By analyzing heterogeneity in the Ct& Int domains, can the authors correlate Ct-Int conformation with the resolution and/or docking position of the Nt domain? Along these lines, can the authors speculate how ALDH1L1 might work as a tetrameric enzyme? Are ALDH1L1 subunits activated stochastically and asynchronously, or is there any coordination among subunits?

3. It seems that the authors have focused on 3D structure classification and masking to improve local resolution in structures such as the N-terminal domain. But recent cryo-EM processing methods based on continuous distributions of electron density (such as cryoDRGN, 3D variability analysis in cryosparc, and e2gmm) are able to improve resolution in flexible domains and parse out more conformational states in many cases. I wonder if the ALDH1L1 structure would benefit from the use of such tools to achieve higher resolution, and/or understand flexibility in ALDH1L1 subdomains.

4. Could the authors validate the subdomain interactions via other methods? For example, if the Nt domain interacts with the Int-Ct core at specific positions, crosslinking mass spectrometry should be able to show the specific proximity of residues. Or performing negative stain EM on chemically crosslinked ALDH1L1 might enrich conformations where transient docking occurs among domains.

5. The authors speculated that the transient interaction of Nt with the NADP+ binding region of Ct

ALDH domains might interfere with NADP⁺ binding. Could the authors measure and compare the NADP⁺ binding affinity of the Ct domain vs. the full-length protein to support this idea?

6. Are structure-function studies possible, where potentially important residues identified in the structure are mutated, and analyzed for function? For example, if the acidity of the Int-Ct linker is important, do neutralizing mutations disrupt the enzyme activity of ALDH1L1?

I understand that this is a long list, and there are many potential difficulties with mutational and functional studies, but I believe that the manuscript would benefit greatly from exploring any one of these avenues.

Minor quibbles are as follows:

1) The title "Structural insight into the function of putative tumor suppressor ALDH1L1" perhaps seems inappropriate as a) the core function of ALDH1L as a metabolic enzyme is known, b) ALDH1L1's role as a tumor suppressor is important, yet peripheral to its core function, and c) the ALDH1L1 structure itself does not lend any insight into its function as a tumor suppressor. A more pared down title true to ALDH1L1's core function might be appropriate.

2) In Fig. 1c, it might be informative if the authors could visualize or label the rough positions of active sites and cofactor binding sites.

3) In Fig. 2d,&e, the structural model in light blue/pink, and the NADP⁺ molecules are a little hard to distinguish. Perhaps a larger figure and/or increased transparency in the electron density map could help make a clearer figure.

4) In Fig. 5e, the point that the Int-Ct domain interaction is generally conserved with respect to other A/PCP-A/PCP partner interactions is well visualized, but the broadness of the contact area is not conveyed well. Perhaps an alternate visualization – more zoomed in and/or with the interaction partner as a surface representation? might help, or coloring residues that are within a specific distance from the A/PCP interaction partner might better convey the idea?

5) Could the authors comment on the distance between the Nt domain active site and the 4'-PP arm to get an idea of how/what distance the 4'-PP arm might move if transporting substrate between the Nt & Ct domains? In line with this, the paper might be more informative and digestible to a broader audience if it had a cartoon schematic showing the proposed mechanistic model of the enzyme specifically showing the position of the 4'-PP arm in the active and inactive states.

6) This was also mentioned in major points, but a discussion of the significance of the tetrameric conformation of ALDH1L1 for enzyme activity would be insightful. Also, a mention of any parallels with other oligomeric/filament-forming metabolic enzymes would help broaden the impact of the paper to a larger community.

Reviewer #4 (Remarks to the Author):

This manuscript describes moderate resolution single-particle cryo-EM analysis of a large multidomain enzyme involved in folate metabolism. The complex multidomain organization of this enzyme has proven to be challenging for structural biology, and crystal structures of only the individual domains have been resolved. The cryo-EM analysis reported here provides the first glimpse of the overall 3D architecture of the full-length enzyme. Although the N-terminal domains are poorly resolved, their structures represent a significant breakthrough and provide a basis for a new chapter in ALDH1L1 biochemistry

Suggestions to make the manuscript clearer:

Figure 1a. The chemical structure of 10-formylTHF is incorrect. As drawn, it has an acetyl group (-

CCH3O) connected to the N atom. It should be formyl (-CHO).

Figure 1a. Use a different color (e.g. red) for the formyl group to show how it moves through the reaction (like in Krupenko, PMID: 18848533).

Figure 1a. The reaction diagram suggests that formic acid is transferred from the N-terminal to the C-terminal domain via the Int domain. In fact, a formyl group is transferred via the 4PP arm.

Figure 1a. The ALDH reaction has the wrong substrate (should be formaldehyde, not formic acid) and is missing both a proton and a water molecule.

Figure 1b. The gaps between domains implies that the enzyme is a complex of three different enzymes, rather than a single enzyme with three covalently connected domains. Remove the gaps between the domains. Again, see Krupenko, PMID: 18848533, Fig. 5.

p. 4, line 84: The authors write, "Overall, in the ALDH1L1 catalysis, the 4'-PP arm of the Int domain transfers the formyl group hydrolytically cleaved from 10-fTHF in the folate binding Nt domain to the Ct domain..." Hydrolytic cleavage of the formyl group presumably produces formic acid (HCO₂H). However, it is the formyl group (-CHO) that is transferred by the 4'-PP arm, so what happened to the O atom from the hydrolytic water molecule?

p. 6. Line 153, The authors write, "Only the adenine portion of bound NADP⁺ was clearly visible in the 153 cryo-EM map, while the nicotinamide moiety of the molecule was disordered." Adenine is a base, as is nicotinamide. Do they mean that only the adenine has density? Perhaps they really mean that the adenosine, AMP, or ADP has density, and the rest of the cofactor is disordered. This is the usual case for disordered NADs bound to ALDHs. The authors could look at the work of the Hurley and Tanner groups on disordered NADs and reference their papers and structures. The references in the current manuscript are dated.

The authors need to show electron density for all the 4'-PPs and all the NADPs as supplementary figures.

The authors would like to thank the reviewers for carefully reading the manuscript and providing detailed and thoughtful comments on presented work. We greatly appreciate the reviewers' recognition of the novelty and significance of our research, and of broad readership interest to the structure of this complex multidomain enzyme with high inter-domain flexibility. As can be seen from the revised manuscript and our responses below, we have addressed the reviewers' concerns by performing additional experiments, including missing methodological details, and discussing the mechanistic, structural and evolutionary aspects of ALDH1L1. Responses to individual comments of each reviewer can be found below.

Reviewer #1 (Remarks to the Author):

1) The source of the enzyme ALDH1L1 that is being studied is not mentioned in the main body of the manuscript and should be included in the first paragraph of results.

We included details on the protein source (recombinant baculovirus expression) and species (*Rattus norvegicus*) in the first sentence of the Results section (lines 102-103): “(*Rattus norvegicus* ALDH1L1 produced in insect cells using a baculovirus expression system)”.

2) Further discussion and speculation of certain aspects would add to the paper. Is there any previous reports on the substrate specificity of ALDH1L1, can it carry out ALDH dehydrogenase activity independent of its other domains? Does the Int domain play a role in stopping spontaneous ALDH activity independent of the other two domains?

Although the substrate specificity of ALDH1L1 was described in the second paragraph of the Introduction in the original manuscript, we did not state explicitly that the N-terminal and C-terminal domains are active as independent units (the former hydrolyses 10-fTHF to THF and formate; the latter converts short-chain aldehydes to corresponding acids as a typical ALDH enzyme). To remedy this, we extended the first paragraph of the Discussion to make it clear that each domain can perform its catalytic reaction independently. Lines 311-317: “Of note, in vitro both the N_t and C_t domains, either expressed individually or within the full-length enzyme, are capable of independent catalysis, 10-formylTHF hydrolysis or small chain aldehyde oxidation, respectively (26, 34). It is not clear whether such independent activities take place in the cell since the hydrolase catalysis in vitro requires high concentrations of non-physiological sulfhydryls while putative substrates for the ALDH reaction are unknown. Thus, the complex mechanism enabled by the merging of the three domains is likely the only catalytic function of ALDH1L1.” All three domains must be present and active for the dehydrogenase activity of ALDH1L1.

The presence of the Int domain does not stop spontaneous ALDH activity (e.g., Donato et al. J Biol Chem. 2007 Nov 23;282(47):34159-66. doi: 10.1074/jbc.M707627200.). The reason for this has not been experimentally determined. We hypothesize that substrate entry is not the rate-limiting step for the ALDH reaction. As an example, in the related ALDH1 protein the reaction rate is limited by the dissociation of the cofactor (Blackwell, L. F., Motion, R. L., MacGibbon, A. K., Hardman, M. J., and Buckley, P. D. (1987) Biochem. J. 242, 803–808). It is therefore plausible that the fraction of the C-terminal protomers with vacant substrate entrance tunnels (estimated to be ~20% in the present manuscript; see “Int domains are highly mobile when not docked into C_t domains”) is sufficient to maintain the rate of the ALDH reaction.

3) The paper lacks comparison to other ALDH members. Discussion of other ALDHs that form tetramers in the same cross-over manner as the Int and Ct domains, e.g. ALDH from *Thermus thermophilus*.

We extended the second paragraph of the Discussion to include comparisons with other ALDH, including ALDH from *Thermus thermophilus*. Lines 331-339: "Of note, although multiple studies analyzed the oligomeric state of ALDH proteins (recently reviewed in (51)), the physiological significance of oligomerization is unclear for most of these enzymes. One possible exception is tetrameric ALDH from *Thermus thermophilus*, which has a ~30 amino acid-long C-terminal extension that interacts with the N-terminal region of a protomer from a different homodimer (52). Other ALDHs, including fatty aldehyde dehydrogenase (FALDH) and ALDH7A1, also feature short C-terminal extensions, but they interact with the protomer within the same homodimer (53, 54). In contrast, ALDH1L1 is the example of an ALDH with two additional domains spanning 400 amino acids at the N-terminus of the enzyme, with the tetrameric state being a prerequisite for its complex function."

4) Other ALDHs that use structural extensions for active site regulation should be discussed, e.g. ALDH7A1, FALDH with a gatekeeper (both from human) and ALDH from *T. thermophilus*.

Discussion of these ALDH family members has been integrated with the response to the previous comment (lines 331-339; see above).

5) It should be proposed whether the Int domain docking in the active site of ALDH potentially plays a role in active site regulation for entry of substrate in ALDH1L1.

We have added corresponding discussion (lines 370-373 of the revised manuscript): "It could also be hypothesized that maintaining the 4'-PP arm within the substrate entrance tunnel prevents the entrance of small aldehydes into the ALDH catalytic center in vivo, thus preserving the enzyme for the 10-formylTHF dehydrogenase catalysis."

6) Comment on the role of oligomerization of the Int and Nt domains in more detail.

The N_t active site is very open, allowing multiple potential modes of interaction with the intermediate domain and 4'-PP (Horita et al. *Chem Biol Interact.* 2017;276:23-30. doi: 10.1016/j.cbi.2017.04.011), which at present makes the prediction of specific conformations rather unreliable. In the original submission we indicated that the ALDH1L1 conformation with the N_t domain resting on the Int-C_t linker might represent a scaffold for N_t-Int interaction. We now added a sentence to the Discussion stating that large domain motions would be necessary for the formation of the N_t-Int complex starting from this conformation (Lines 394-396: "In the latter scenario, a large-scale rotation and shift of the Int domain would be necessary to bring together the sulfhydryl of the 4'-PP arm and the N_t active site residues, which are about 50 Å apart.>").

7) Has the expressed and purified full length protein been trialled for enzymatic activity? Can the conversion of substrate to product be obtained with a mixture of individual domains in solution or is the combination of domains and tetrameric assembly essential.

Upon purification, protein samples were tested for purity, and all three enzymatic activities were assayed. A corresponding statement has been added to the Methods (lines 440-444): "Purified full-length ALDH1L1 was tested for the 10-formylTHF dehydrogenase activity as we previously described (33). C_t domain and Int-C_t protein were tested for the aldehyde dehydrogenase activity using propanal as

the substrate and NADP⁺ as the cofactor essentially as we described (26). After purification, all protein preparations used in the present study had specific activities close to previously reported values (26, 33) ...”

It is currently unknown whether mixing the three domains would produce the dehydrogenase activity (an expression system for the individual intermediate domain that would result in post-translational modification of Ser354 with 4'-PP has not been established). We hypothesize that very high concentrations of the N-terminal and intermediate domains would be required to reproduce their effective “concentrations” in full-length ALDH1L1.

8) The reviewer suggests making a more distinct colour contrast being domains for figure 2d, 2e and 7 to make visualisation clearer. For figure 3c, the subdomains of the ALDH domain could be coloured differently rather than just labelled.

We followed the suggestion of the reviewer to improve the contrast in panels d and e of Figure 2. We altered the colors in Fig. 7 for better visualization and illustrated the linker between the N_t and C_t domain as a spring. Finally, we colored the sub-domains of the ALDH domain individually in Figure 3, panel c. Please see the revised figures 2, 7 and 3 below.

Fig. 2. The architecture of ALDH1L1 includes a rigid core and highly mobile N-terminal domains. **a:** Comparison of negative-stain EM 2D class averages of the individually expressed tetrameric C-terminal domain (C_t, *left* panel) and the full-length protein (*right* panel) was used for identification of intermediate domains (Int) adjacent to the C_t domain within each protomer. Only a single N-terminal domain (N_t) is clearly visible. **b:** Selected negative-stain 2D classes illustrate the range of positions assumed by the N_t domains within the ALDH1L1 tetramer. Arrows point to the position of the N_t domain in which it closely interacts with the C_t domain. **c:** The range of motion of the N_t domain illustrated by a selection of negative-stain 2D classes. Below: cartoon representations of the crystal structure of N_t in the corresponding orientations. **d, e:** Cryo-EM structures of ligand-free ALDH1L1 (**d**) and ALDH1L1 in complex with NADP⁺ (**e**). The maps are displayed as transparent surfaces. Atomic models are shown in *cartoon* representation, with Int and C_t domains colored *light-blue* and *pink*, respectively. Bound NADP⁺ molecules are shown in *sphere* representation. **f:** Cryo-EM structure of the rigid core of ALDH1L1 corresponds well to the negative-stain EM 2D classes. Results of segmentation of the ligand-free map are shown. Scale bars in panels **a-c** correspond to 10 nm.

Fig. 7. Transient interactions of the N-terminal (N_t) domain of ALDH1L1. **a:** Relaxation of the D2 symmetry led to appearance of weak density for one N_t domain in the cryo-EM map of ligand-free ALDH1L1. **b:** Conformation with two symmetrically located N_t domains. The N_t domains interact with the regions of the C_t core responsible for binding of $NADP^+$. **c:** Conformation of ALDH1L1 with one N_t domain interacting with the linker between the intermediate and the C_t domains (Int- C_t linker) of a different protomer. The linker is represented with the black spring. The 4'-PP arms are in sphere representation. In the two images to the right, the surface of the N_t domain is colored according to the electrostatic potential. The positively charged patch interacting with the Int- C_t linker is enclosed in *dashed ovals*. In the remaining images, the N_t , Int, and C_t domains are colored *gold*, *blue*, and *salmon*, respectively.

Fig. 3. Structures of the intermediate and C_t domains of ALDH1L1. **a:** Angular views of the intermediate domain. The molecular model is shown as *sticks*, and the cryo-EM density is represented by semi-transparent surface. 4PP: 4'-phosphopantetheine. **b:** Superposition of the intermediate domain of ALDH1L1 (*light blue*) and the carrier domain of holo-non-ribosomal peptidyl synthetase (PDB 4zxh, *grey*). The alpha-helices are labeled I to IV. The r.m.s.d. between the two structures is 1.98 Å. **c:** Structure of the protomer of the C-terminal domain. The three sub-domains are colored individually and labeled. Bound $NADP^+$ is shown in *sphere* representation. **d:** Superposition of the C_t domains of ligand free (*grey*) and $NADP^+$ -bound (*light blue*) ALDH1L1. Binding of $NADP^+$ induces local structural changes in one helix forming the binding cleft for the adenine moiety (*dashed circle*). Cryo-EM density for $NADP^+$ is shown as semi-transparent surface.

Other minor corrections:

Line 192-198: Should this structural superimposition refer to a figure

Since the structural comparisons mentioned in this part of the text did not find significant differences, we believe it would be impractical to dedicate a figure to the description of these results.

Line 390: 30 μg not thirty μg

We corrected the text as suggested (line 447), although we believe numbers are typically spelled out in the beginning of the sentence.

Reviewer #2 (Remarks to the Author):

1) They found a disulfide bond between the catalytic Cys of the ALDH domain and the sulfur of the 4'-PP prosthetic group in a population of the enzyme molecules and hypothesize that formation of this covalent bond may be a mechanism of protection against oxidation of the highly reactive thiols of the catalytic cysteine and 4'-PP in the absence of the substrate. However, the formation of this disulfide would irreversibly inactivate the ALDH and sequester 4'-PP, so it does not provide any protection. Different from a thiol-nicotinamide adduct, which is reversible, a disulfide bond can only be broken by disulfide thiol exchange, and it is difficult to envision that physiological thiols, such as reduced glutathione for instance, could reach the position of this disulfide deep into the ALDH active-site tunnel. In my opinion, this disulfide is indeed the result of the oxidation of the two thiols during the experiments. In the text there is no mention to the inclusion of a reducing agent in the buffers used during the purification of the protein, its storage or in the protein solution used in the Cry-EM experiments. Perhaps if such compounds were included in the protein solution, and/or if the Cry-EM experiments were carried under anaerobic conditions if possible, the disulfide would have not been formed.

We agree with the reviewer regarding the potential inactivation of the enzyme through the observed disulfide bond formation and re-thought our interpretation of this phenomenon. We now state that (a) the disulfide could be a result of spontaneous oxidation in our experiments setting or (b) the bond could still be reduced in the presence of natural reducing agents such as cysteine or glutathione. These reducing agents can reach the catalytic cysteine through the large NADP binding site located on the side of the ALDH protomer opposite to the substrate entrance tunnel. The corresponding section of the Discussion (paragraph 4) has been modified to reflect these interpretations of the disulfide bond. Lines 367-374: "This disulfide could be reduced by cellular glutathione accessing the active center through the NADP⁺ binding site. Of note, in the individually expressed C_t domain Cys707 was shown to form a transient covalent adduct with the C4 atom of the nicotinamide ring of NADP⁺ (27), which suggests that this cysteine is highly reactive beyond the immediate catalytic step. It could also be hypothesized that maintaining the 4'-PP arm within the substrate entrance tunnel prevents the entrance of small aldehydes into the ALDH catalytic center in vivo, thus preserving the enzyme for the 10-formylTHF dehydrogenase catalysis. Alternatively, we cannot exclude the possibility that the observed disulfide bond is the result of oxidation in our experimental setting." Furthermore, the expanded "Protein expression and purification" section in the revised Methods also indicates that a reducing agent was present during ALDH1L1 purification and storage.

Lines 425-427: "To purify ALDH1L1, the cell culture medium was applied to a column containing 5-formyl-THF-Sepharose affinity resin equilibrated with 10 mM Tris-HCl buffer, pH 7.4, containing 10 mM 2-ME and 1 mM NaN₃ (buffer A)."

Lines 443-445: "After purification, all protein preparations used in the present study had specific activities close to previously reported values (26, 33) and were stored at -80°C in the presence of 10 mM 2-ME and 20% glycerol."

2)Fig. 1: The formyl group in the 10-formyltetrahydrofolate formula should have a hydrogen attached to the carbonyl C=O; delete the line, or substitute it for H, so that the aldehyde group is clearly seen. Formic acid is not formed as an intermediate of the reaction. Show the correct formaldehyde intermediate and the 4'-PP bound to this intermediate.

We are grateful to the reviewer for noticing this error. It has been corrected in revised Fig. 1, shown below.

Fig. 1. Domain organization and catalytic function of 10-formyltetrahydrofolate dehydrogenase (ALDH1L1). **a:** ALDH1L1 converts 10-formyltetrahydrofolate (10-formylTHF) to tetrahydrofolate (THF) and CO₂ in two sequential enzymatic reactions, the hydrolytic cleavage of the formyl group in the folate binding N-terminal domain and the NADP⁺-dependent oxidation of the formyl to carbon dioxide in the aldehyde dehydrogenase C-terminal domain. The intermediate domain, which is homologous of acyl/peptidyl carrier proteins, transports the formyl group, attached to the 4'-phosphopantetheine prosthetic arm (4PP), from the N-terminal to the C-terminal domain. **b:** Primary structure of ALDH1L1 with indicated domain boundaries. **c:** Schematic of spatial organization of the ALDH1L1 tetramer based on available atomic structures of its individual domains. PDB structures 1s3i, 2cq8, and 2o2p were used for the N-terminal, intermediate, and C-terminal domains, respectively. For one protomer the domains are labeled and colored as in **b**. Arrows mark the positions of the hydrolase and aldehyde dehydrogenase active sites (AS) and the NADP⁺ binding site.

Minor points:

1) Introduction, line 79: 4'-PP is bound through a phosphoester bond, not by a phosphodiester as said.

We corrected the text as suggested (line 78).

2) Results, line 191: 'the amino group of Arg359' should be 'the guanidine (or guanidinium) group of Arg359'.

We corrected the text as suggested (line 191).

3) Give a more detailed description of the protein purification method; the work cited as this respect does not give such details.

We have significantly extended the "Protein expression and purification" section in the Methods to address this comment (lines 420-445).

Reviewer #3

Some of my major suggestions are summarized as follows:

1. The authors suggest that the cryo-EM structure reflects the "resting state" of the enzyme. This is plausible considering that the Int domain & 4'-PP is plugging the substrate entrance tunnel all the way to the catalytic Cys without any substrate. However, the burning question in my mind is whether the full-length enzyme used in the structural studies has any enzymatic (hydrolase/dehydrogenase) activity, i.e., is the ALDH1L1 used in the study competent to assume an active state upon addition of substrate? If so, is there any gross structural change in ALDH1L1 upon substrate addition? If not, why is the enzyme inactive, and what other factors are required to transition ALDH1L1 to an active state?

The ALDH1L1 used in this work was tested for 10-formylTHF dehydrogenase activity after purification, and it was found to be active (a statement reflecting this has been added to the Methods, lines 440-444): "Purified full-length ALDH1L1 was tested for the 10-formylTHF dehydrogenase activity as we previously described (33). Ct domain and Int-Ct protein were tested for the aldehyde dehydrogenase activity using propanal as the substrate and NADP⁺ as the cofactor essentially as we described (26). After purification, all protein preparations used in the present study had specific activities close to previously reported values (26, 33) ..." This indicates that both catalytic centers of the enzyme were functional, and the protein was competent to assume an active state upon addition of substrate and cofactor. During the preliminary phase of the structural analysis, we investigated how addition of a substrate analog 10-formylidideazafolate and NADP⁺ affected the conformation of ALDH1L1 using negative-stain EM. There were no major differences, but we noticed that the 2D classes with no intermediate domains docked into the substrate entrance tunnels of the ALDH core appeared more commonly in these experiments (see the figure below for examples; arrows point to absent Int domains). Our interpretation was that in a "working" enzyme the intermediate domains tend to spend less time interacting with the C_i core due to the need to shuttle the formyl group from the N_i domains. This observation motivated us to choose the "resting state" for cryo-EM studies to reduce the conformational heterogeneity of the protein. This is

just a qualitative observation; quantification of the occupancy of the Int domain would likely be of limited accuracy because of its small size and limited resolution of the NS-EM method.

2. The authors impose D2 symmetry on the cryo-EM data processing to get to a higher-resolution structure, which itself is a fully valid method, but I wonder whether valuable information is being lost by imposing symmetry. When relaxing or omitting the symmetry constraints, do the authors find any differences among subunits in the Ct & Int domains? By analyzing heterogeneity in the Ct& Int domains, can the authors correlate Ct-Int conformation with the resolution and/or docking position of the Nt domain? Along these lines, can the authors speculate how ALDH1L1 might work as a tetrameric enzyme? Are ALDH1L1 subunits activated stochastically and asynchronously, or is there any coordination among subunits?

In the original submission, we indicated that the structure of ALDH1L1 refined without symmetry constraints (4.4 Å resolution) was highly similar to the symmetrical structure (first paragraph of the subsection of the Results titled “Transient interactions of the N-terminal domain”; lines 261-262 of the revised manuscript). To verify the possibility of structural differences between the protomers of this structure induced by the presence of density for one N_t domain, we calculated root mean square deviations (r.m.s.ds) between all pairs of chains for this non-symmetrical structure and found them to be very small (0.67 to 0.74 Å). We conclude that the ALDH1L1 core remains rigid, which agrees with what we found previously for the ALDH domain using X-ray crystallography (Tsybovsky et al. *Biochemistry*. 2007;46(11):2917-29; Tsybovsky, Krupenko. *J Biol Chem*. 2011;286(26):23357-67). We have included this finding in the revised manuscript (p. 264-265): “Low r.m.s.d. values between the chains of this structure (0.67-0.74 Å) indicated that this interaction did not induce large structural rearrangements in the protein core.” Unfortunately, the resolution of the remaining two structures with visible N_t domain(s) (~7 Å) does not allow reliable model building and hence comparison of the protomers. Our current view is that the ALDH1L1 catalysis is primarily driven by stochastic domains movements with transient interactions contributing to the catalytic mechanism. To make this clear, we have modified corresponding statements in the revised manuscript as outlined below.

Lines 375-376: “The high mobility of the N_t domains suggests that ALDH1L1 catalysis is driven primarily by stochastic domain movements.”

Lines 396-399: “While the above interpretations are speculative, the existence of scarcely populated states with firmly positioned hydrolase domains alludes to an intricate mechanism of catalysis that may involve various auxiliary inter-domain interactions guiding the overall random domain movements during catalysis.”

3. It seems that the authors have focused on 3D structure classification and masking to improve local

resolution in structures such as the N-terminal domain. But recent cryo-EM processing methods based on continuous distributions of electron density (such as cryoDRGN, 3D variability analysis in cryosparc, and e2gmm) are able to improve resolution in flexible domains and parse out more conformational states in many cases. I wonder if the ALDH1L1 structure would benefit from the use of such tools to achieve higher resolution, and/or understand flexibility in ALDH1L1 subdomains.

We agree with the reviewer that the recently introduced computational tools for detecting and visualizing conformational heterogeneity in cryo-EM data can be very beneficial in many cases, which is exemplified by several recent publications. For instance, the first author of this manuscript utilized 3D variability analysis in cryoSPARC to resolve domain movements of the SARS-CoV-2 spike glycoprotein (Zhou, Tsybovsky et al. *Cell Host Microbe*. 2020 Dec 9;28(6):867-879.e5. doi: 10.1016/j.chom.2020.11.004). In the course of structural analysis of ALDH1L1, we attempted to use this methodology first to resolve the N-terminal domains but did not succeed. Specifically, the 3D variability analysis produced structures with varying strength of the N_t density rather than with different N_t positions. We believe that the main reasons for this are a very high magnitude of movements (as demonstrated by negative-stain EM, Fig. 3b,c) combined with the small size of the moving part by cryo-EM standards. The problem was likely exacerbated by the presence of four N-terminal domains that moved simultaneously and independently from each other. Based on this outcome, we concluded that resolving the movements of the N-terminal domains of ALDH1L1 is beyond the capabilities of the current 3D variability approaches. Therefore, we had to resort to using 3D classification to isolate “stable” conformational states and on 2D classification of NS-EM data to demonstrate the magnitude of domain movements.

4. Could the authors validate the subdomain interactions via other methods? For example, if the N_t domain interacts with the Int-Ct core at specific positions, crosslinking mass spectrometry should be able to show the specific proximity of residues. Or performing negative stain EM on chemically crosslinked ALDH1L1 might enrich conformations where transient docking occurs among domains.

As the reviewer suggested, we performed chemical cross-linking of full-length ALDH1L1 with glutaraldehyde coupled with negative-stain electron microscopy. We found that glutaraldehyde treatment gradually reduced the fraction of ALDH1L1 molecules with visible “free” arms (N-terminal domains). Simultaneously, new 2D classes emerged illustrating N-terminal domains in contact with the protein core. This confirms that the N-terminal domains form transient interactions with the ALDH1L1 core at specific positions. The results of this experiment are described in the last paragraph of the Results section in the revised manuscript (lines 284-291): “To confirm the formation of transient complexes between N_t domains and the ALDH1L1 core, we performed chemical cross-linking of the full-length protein with 0.1% glutaraldehyde followed by NS-EM and 2D classification (Fig. S3a). This treatment resulted in gradual disappearance of 2D classes displaying N_t moieties not in contact with the protein core (Fig. S3b), indicating that glutaraldehyde cross-linking stabilized the transient complexes formed by the N-terminal domains. Of note, N_t domains attached to the core were reliably resolved in the 2D class averages even after prolonged glutaraldehyde treatment, suggesting that cross-linking occurred at specific positions.” Additionally, they are illustrated in a new supplementary figure (revised Fig. S3):

a

b

Supplementary Fig. 3. Cross-linking with glutaraldehyde immobilizes N-terminal domains of ALDH1L1 in non-random positions. **a:** Negative-stain 2D class averages of ALDH1L1 treated with 0.1% glutaraldehyde at 4°C for 0-60 min. Arrows point to visible mobile "arms" (N-terminal domains). **b:** Results of quantification of resolved mobile arms versus incubation time.

The description of the methodology was added to the Methods under “Chemical cross-linking and comparative quantification of mobile N-terminal domains” (lines 480-492).

5. The authors speculated that the transient interaction of Nt with the NADP⁺ binding region of Ct ALDH domains might interfere with NADP⁺ binding. Could the authors measure and compare the NADP⁺ binding affinity of the Ct domain vs. the full-length protein to support this idea?

The NADP⁺ binding affinities of the C_t domain and full-length ALDH1L1 were compared previously (Krupenko et al., JBC, 1997, Volume 272, Issue 15, Pages 10266-10272, doi: 10.1074/jbc.272.15.10266) and found to be similar (0.2 vs. 0.3 μM, respectively). This indicates that the proposed role of the hydrolase domains in regulating NADP⁺ binding is likely small. This is in line with the small size of the corresponding cryo-EM sub-dataset in the present manuscript (5,517 particles vs. 86,276 particles in the full dataset). To make this clear to the readers, we explicitly stated the magnitude of the proposed effect in the paragraph of the Discussion describing the transient complexes of the N_t domain (lines 384-387): “It has to be noted, however, that full-length ALDH1L1 and the individually expressed ALDH domain displayed similar affinities for NADP (K_d of 0.3 μM versus 0.2 μM, respectively) (26), suggesting that the proposed effect is likely small.” We thank the reviewer for the opportunity to provide more accurate conclusions.

6. Are structure-function studies possible, where potentially important residues identified in the structure are mutated, and analyzed for function? For example, if the acidity of the Int-Ct linker is important, do neutralizing mutations disrupt the enzyme activity of ALDH1L1?

I understand that this is a long list, and there are many potential difficulties with mutational and functional studies, but I believe that the manuscript would benefit greatly from exploring any one of these avenues.

The role of the interdomain linker region in ALDH1L1 catalysis was probed by site-directed mutagenesis in our previous study (Reuland et al., JBC, 2003, Volume 278, Issue 25, pp. 22894-900, doi: 10.1074/jbc.M302948200). Specifically, the first three residues of the linker, including negatively charged Glu398, as well as the three preceding residues, were mutated to alanines individually and simultaneously. While single substitutions did not affect the activity of ALDH1L1, the replacement of all six amino acids resulted in essentially inactive enzyme, which was at the time ascribed to the loss of flexibility in this region. These results assert that the conformation of the linker is important for function and illustrate a common observation that single mutations rarely dramatically influence protein-protein interactions. While we agree with the reviewer that mutational analysis of the linker could be informative, we also concur that such studies have multiple potential difficulties. We feel that more comprehensive mutagenesis analysis of the entire linker is outside of the scope of the present study and hope that the reviewer will agree.

Minor quibbles are as follows:

1) The title “Structural insight into the function of putative tumor suppressor ALDH1L1” perhaps seems inappropriate as a) the core function of ALDH1L as a metabolic enzyme is known, b) ALDH1L1’s role as a tumor suppressor is important, yet peripheral to its core function, and c) the ALDH1L1 structure itself does not lend any insight into its function as a tumor suppressor. A more pared down title true to ALDH1L1’s core function might be appropriate.

We agree that the cryo-EM structures do not lend insight into the tumor suppressor function of ALDH1L1. Therefore, we have changed the title of the manuscript to “Structure of putative tumor suppressor ALDH1L1.”

2) In Fig. 1c, it might be informative if the authors could visualize or label the rough positions of active sites and cofactor binding sites.

As suggested by the reviewer, we marked the approximate locations of the active sites of the N-terminal and C-terminal domains, as well as the position of the NADP binding site, in revised Fig. 1c (below).

Fig. 1. Domain organization and catalytic function of 10-formyltetrahydrofolate dehydrogenase (ALDH1L1). **a:** ALDH1L1 converts 10-formyltetrahydrofolate (10-formylTHF) to tetrahydrofolate (THF) and CO_2 in two sequential enzymatic reactions, the hydrolytic cleavage of the formyl group in the folate binding N-terminal domain and the NADP^+ -dependent oxidation of the formyl to carbon dioxide in the aldehyde dehydrogenase C-terminal domain. The intermediate domain, which is homologous of acyl/peptidyl carrier proteins, transports the formyl group, attached to the 4'-phosphopantetheine prosthetic arm (4PP), from the N-terminal to the C-terminal domain. **b:** Primary structure of ALDH1L1 with indicated domain boundaries. **c:** Schematic of spatial organization of the ALDH1L1 tetramer based on available atomic structures of its individual domains. PDB structures 1s3i, 2cq8, and 2o2p were used for the N-terminal, intermediate, and C-terminal domains, respectively. For one protomer the domains are labeled and colored as in **b**. Arrows mark the positions of the hydrolase and aldehyde dehydrogenase active sites (AS) and the NADP^+ binding site.

3) In Fig. 2d,e, the structural model in light blue/pink, and the NADP^+ molecules are a little hard to distinguish. Perhaps a larger figure and/or increased transparency in the electron density map could help make a clearer figure.

We agree with the reviewer. We modified panels d and e of Fig. 2 by increasing the transparency of the cryo-EM density map and slightly enlarging the panels (below).

Fig. 2. The architecture of ALDH1L1 includes a rigid core and highly mobile N-terminal domains. **a:** Comparison of negative-stain EM 2D class averages of the individually expressed tetrameric C-terminal domain (C_t , *left* panel) and the full-length protein (*right* panel) was used for identification of intermediate domains (Int) adjacent to the C_t domain within each protomer. Only a single N-terminal domain (N_t) is clearly visible. **b:** Selected negative-stain 2D classes illustrate the range of positions assumed by the N_t domains within the ALDH1L1 tetramer. Arrows point to the position of the N_t domain in which it closely interacts with the C_t domain. **c:** The range of motion of the N_t domain illustrated by a selection of negative-stain 2D classes. Below: cartoon representations of the crystal structure of N_t in the corresponding orientations. **d, e:** Cryo-EM structures of ligand-free ALDH1L1 (**d**) and ALDH1L1 in complex with NADP⁺ (**e**). The maps are displayed as transparent surfaces. Atomic models are shown in *cartoon* representation, with Int and C_t domains colored *light-blue* and *pink*, respectively. Bound NADP⁺ molecules are shown in *sphere* representation. **f:** Cryo-EM structure of the rigid core of ALDH1L1 corresponds well to the negative-stain EM 2D classes. Results of segmentation of the ligand-free map are shown. Scale bars in panels **a-c** correspond to 10 nm.

4) In Fig. 5e, the point that the Int- C_t domain interaction is generally conserved with respect to other A/PCP-A/PCP partner interactions is well visualized, but the broadness of the contact area is not conveyed well. Perhaps an alternate visualization – more zoomed in and/or with the interaction partner as a surface representation? might help, or coloring residues that are within a specific distance from the A/PCP interaction partner might better convey the idea?

To address this comment, for each complex in Fig. 5e we identified the residues in contact with A/PCP protein using the UCSF Chimera “Find Contacts” function. The side chains of these residues are shown in sphere representation in the revised figure. This allows better visualization of the broadness of the contact area of the Int- C_t domain interaction (see the revised figure below).

Fig. 5. Interactions between the intermediate (Int) and C-terminal (C_t) domains of ALDH1D1. **a:** The intermediate domain (light blue) depicted in *pipes-and-planks* representation contacts C_t domains of two protomers (pink and cyan). Only the oligomerization sub-domain is shown for the cyan protomer. The 4'-phosphopantetheine (4PP) arm is shown in *stick* representation. The alpha-helices of the Int domain are labeled I to IV. **b:** Contacting surfaces of the Int domain and the two C_t domains. Coloring is according to panel a. *Left*, surface representations of the complexes. *Right*, the complexes are split, and the domains are shown with the interfaces towards the viewer. Contact surfaces are colored according to the color of the interacting partner. **c:** Surfaces of the Int and C_t domains colored according to the electrostatic potentials, with red and blue corresponding to negative and positive charge, respectively. *Left*: the domains are shown in the same orientations as in the complex. The arrow points to the substrate entrance tunnel. *Right*: The same domains are rotated such that the contact interfaces face the viewer. **d:** A comparison of negative-stain 2D classes of individual C_t domain and the Int- C_t protein expressed in bacteria. *White arrows* point to Int domains adjacent to the C_t core. *Red arrows* point to vacant sites for binding of Int domain. Scale bars represent 10 nm. **e:** A comparison of the complex between the Int and C_t domains of ALDH1L1 with various complexes formed by acyl and peptidyl carrier proteins. The carrier proteins are aligned to the Int domain. The loop connecting helices I and II is colored orange. The residues contacting A/PCPs are shown in sphere representation. The PDB code is listed below each complex. 1f80: complex between *Bacillus subtilis* ACP and ACP synthase; 2cg5: human fatty acid synthase ACP in complex with phosphopantetheinyl transferase; 3ejb: *E. coli* ACP in complex with cytochrome P450; 3ny7: *E. coli* ACP in complex with STAS domain of YchM anion transporter; 4zxh: *Acinetobacter baumannii* PCP in complex with nonribosomal peptide synthetase; 5es8: PCP in complex with adenylation domain of nonribosomal peptide synthase from *Brevibacillus parabrevis*.

5) Could the authors comment on the distance between the Nt domain active site and the 4'-PP arm to get an idea of how/what distance the 4'-PP arm might move if transporting substrate between the Nt & Ct domains? In line with this, the paper might be more informative and digestible to a broader audience

if it had a cartoon schematic showing the proposed mechanistic model of the enzyme specifically showing the position of the 4'-PP arm in the active and inactive states.

The distance between the sulfhydryl of the 4'-PP arm and the active site His106 of the N_t domain in the structure where this domain rests on the Int-C_t linker (Fig. 7c) is about 50 Å. A large rotation and shift of the Int would be necessary to bring the 4'-PP to the N_t active site. We mention this in the Discussion of the revised manuscript (lines 394-396): "In the latter scenario, a large-scale rotation and shift of the Int domain would be necessary to bring together the sulfhydryl of the 4'-PP arm and the N_t active site residues, which are about 50 Å apart." We also added the 4'-PP arms in the sphere representation in Fig. 7c to illustrate their locations with respect to the N-terminal domain. We feel that the proposed cartoon schematic would be rather speculative because there is currently no structural information for the complex between the N-terminal and intermediate domains, and we are reluctant to include it in the manuscript.

6) This was also mentioned in major points, but a discussion of the significance of the tetrameric conformation of ALDH1L1 for enzyme activity would be insightful. Also, a mention of any parallels with other oligomeric/filament-forming metabolic enzymes would help broaden the impact of the paper to a larger community.

Paragraph 2 of the Discussion in the original manuscript included limited information on the biological significance of the tetrameric organization of ALDH1L1. We have now extended it to include the discussion of other ALDH enzymes with sequence extensions. Lines 331-339: "Of note, although multiple studies analyzed the oligomeric state of ALDH proteins (recently reviewed in (51)), the physiological significance of oligomerization is unclear for most of these enzymes. One possible exception is tetrameric ALDH from *Thermus thermophilus*, which has a ~30 amino acid-long C-terminal extension that interacts with the N-terminal region of a protomer from a different homodimer (52). Other ALDHs, including fatty aldehyde dehydrogenase (FALDH) and ALDH7A1, also feature short C-terminal extensions, but they interact with the protomer within the same homodimer (53, 54). In contrast, ALDH1L1 is the example of an ALDH with two additional domains spanning 400 amino acids at the N-terminus of the enzyme, with the tetrameric state being a prerequisite for its complex function."

Based on the available literature, the biological role of oligomerization for most ALDH enzymes is currently unclear. In this regard, ALDH1L1 is a rare example in which the tetrameric state has now been demonstrated to be a prerequisite for enzymatic activity. Unfortunately, we are unaware of oligomeric metabolic enzymes sufficiently structurally similar to ALDH1L1 to warrant mentioning in the manuscript. However, we have included a general discussion of the role of enzyme oligomerization and modular organization in the last paragraph of the Discussion. Lines 402-412: "Protein oligomerization and multidomain organization are common phenomena in eukaryotes (60-63). While modular organization can expand the enzyme functionality (61), oligomerization provides benefits such as efficiency, regulation and stability (63). In some cases, oligomerization is required because catalytic centers are formed by residues from different protomers or because oligomers enable additional non-catalytic regulatory sites. Metabolic enzymes can also form structures of higher degree of order like filaments, which might not directly affect the catalysis within a single unit (64). Of note, all such examples were reported in folate metabolism where ALDH1L1 belongs (65-68). Here we uncovered another mechanism in which tetrameric organization allows modular catalysis bypassing spatial restrictions within a single protomer. Thus, the tetrameric state of ALDH1L1 is indispensable for the enzyme functionality, which also involves transient domain interactions and large-scale domain movements."

Reviewer #4 (Remarks to the Author):

1) Figure 1a. The chemical structure of 10-formylTHF is incorrect. As drawn, it has an acetyl group (-CCH₃O) connected to the N atom. It should be formyl (-CHO).

We are grateful to the reviewer for pointing out this inaccuracy in Fig. 1a, as well as the ones described below. We corrected the figure to include the proper chemical structure of 10-formylTHF, as shown below.

Fig. 1. Domain organization and catalytic function of 10-formyltetrahydrofolate dehydrogenase (ALDH1L1). **a:** ALDH1L1 converts 10-formyltetrahydrofolate (10-formylTHF) to tetrahydrofolate (THF) and CO_2 in two sequential enzymatic reactions, the hydrolytic cleavage of the formyl group in the folate binding N-terminal domain and the NADP^+ -dependent oxidation of the formyl to carbon dioxide in the aldehyde dehydrogenase C-terminal domain. The intermediate domain, which is homologous of acyl/peptidyl carrier proteins, transports the formyl group, attached to the 4'-phosphopantetheine prosthetic arm (4PP), from the N-terminal to the C-terminal domain. **b:** Primary structure of ALDH1L1 with indicated domain boundaries. **c:** Schematic of spatial organization of the ALDH1L1 tetramer based on available atomic structures of its individual domains. PDB structures 1s3i, 2cq8, and 2o2p were used for the N-terminal, intermediate, and C-terminal domains, respectively. For one protomer the domains are labeled and colored as in **b**. Arrows mark the positions of the hydrolase and aldehyde dehydrogenase active sites (AS) and the NADP^+ binding site.

2) Figure 1a. Use a different color (e.g. red) for the formyl group to show how it moves through the reaction (like in Krupenko, PMID: 18848533).

As suggested, we colored the formyl group red in Fig. 1a (see above).

3) Figure 1a. The reaction diagram suggests that formic acid is transferred from the N-terminal to the C-terminal domain via the Int domain. In fact, a formyl group is transferred via the 4PP arm.

This error has been corrected in the revised figure (see above).

4) Figure 1a. The ALDH reaction has the wrong substrate (should be formaldehyde, not formic acid) and is missing both a proton and a water molecule.

These inaccuracies have been corrected in revised figure Fig. 1a (see above).

5) Figure 1b. The gaps between domains implies that the enzyme is a complex of three different enzymes, rather than a single enzyme with three covalently connected domains. Remove the gaps between the domains. Again, see Krupenko, PMID: 18848533, Fig. 5.

Our original intent was to use the gaps to show the linkers between the three domains. We agree with the reviewer that this mode of presentation can be confusing. To remedy this, we filled the gaps with grey rectangles representing the linkers (see Fig. 1 above).

6) p. 4, line 84: The authors write, "Overall, in the ALDH1L1 catalysis, the 4'-PP arm of the Int domain transfers the formyl group hydrolytically cleaved from 10-fTHF in the folate binding Nt domain to the Ct domain..." Hydrolytic cleavage of the formyl group presumably produces formic acid (HCO₂H). However, it is the formyl group (-CHO) that is transferred by the 4'-PP arm, so what happened to the O atom from the hydrolytic water molecule?

To clarify the proposed mechanism of transfer of the formyl to the 4'-PP arm we illustrated it in a new supplementary figure (Fig. S4; referenced in lines 310-311). Overall, although a water molecule participates in the reaction, allowing the discharge of the formyl from the substrate, it is not used up because the formyl group is further transferred to 4'-PP.

Supplementary Fig. 4. Proposed two-step mechanism of ALDH1L1 catalysis. **a:** Formyl transfer from 10-formylTHF to the 4'-phosphopantetheine (4'-PP) arm in the folate-binding hydrolase domain (N_f -domain). Asp142 activates a water molecule for hydrolysis of the bond between the carbonyl carbon of the formyl group and N^{10} of 10-formylTHF while His106 helps to orient the carbonyl group. Formate remains bound in the active site until the 4'-PP arm swings in. The sulfur of the 4'-PP attacks the carbon of the formate, forming a covalent bond, with simultaneous transfer of the proton from the sulfhydryl group to the oxygen, thus releasing a water molecule. Overall, the water molecule that participates in this step is restored upon the transfer of the formyl group to 4'-PP. **b:** Formyl oxidation in the C-terminal ALDH domain (Ct-domain). The formyl group is first transferred from the 4'-PP moiety to the sulfur atom of catalytic Cys707 and then is oxidized to CO_2 in the presence of NADP^+ ; a water molecule activated by Glu673 participates in this step.

7) p. 6. Line 153, The authors write, "Only the adenine portion of bound NADP^+ was clearly visible in the 153 cryo-EM map, while the nicotinamide moiety of the molecule was disordered." Adenine is a base, as is nicotinamide. Do they mean that only the adenine has density? Perhaps they really mean that the adenosine, AMP, or ADP has density, and the rest of the cofactor is disordered. This is the usual case for disordered NADs bound to ALDHs. The authors could look at the work of the Hurley and Tanner groups on disordered NADs and reference their papers and structures. The references in the current manuscript are dated.

The reviewer is correct in pointing out that our terminology was flawed. We corrected it as suggested (lines 153-156): "Only the AMP portion of bound NADP^+ was clearly visible in the cryo-EM map, while there was no density for the rest of the cofactor. Of note, weak density for the nicotinamide riboside of NAD^+ and NADP^+ is commonly observed in aldehyde dehydrogenases (27, 43, 44)." Additionally, we replaced the corresponding references with more recent papers.

8) The authors need to show electron density for all the 4'-PPs and all the NADPs as supplementary figures.

The cryo-EM densities for all the 4'-PP arms and NADP⁺ molecules are included in Fig. S6 of the revised manuscript:

Supplementary Fig. 7. Cryo-EM density for the 4'-phosphopantetheine (4PP) arms and NADP⁺ molecules in the cryo-EM structures of ligand-free ALDH1L1 and ALDH1L1 in complex with NADP⁺. For NADP⁺, only density for the adenine, ribose and two phosphates was present.

This figure is referenced in lines 537-539: "Representative micrographs and 2D class averages, FSC curves, and local resolution data are presented in Fig. S5 and Fig. S6. Fig. S7 illustrates the cryo-EM density for 4'-PP and NADP⁺."

REVIEWERS' COMMENTS:

Reviewer #2 (Remarks to the Author):

In the revised version of their manuscript, Tsybovsky et al. satisfactorily addressed my recommendations. Therefore, in my opinion, this revised version could be accepted for publication.

Reviewer #3 (Remarks to the Author):

I would like to thank the authors for providing a comprehensive response to the questions and comments raised about the original submitted manuscript.

The major suggestions that were made in my initial review and the authors' response were as follows:

1. Clarification of the structural state of the enzyme, i.e., why the authors chose to interpret the enzyme as the "resting state", and if the enzyme used for structure determination has any activity.
> In the Methods section, the authors clarified that the purified ALDH1L1 possesses enzymatic activity, but that "the resting state" without substrate was more amenable to structure determination.
>> The comment has resolved a major curiosity that was unanswered in the original submission.

2. A more detailed analysis of different conformational states among protomers, and application of alternative data processing tools to increase resolution and/or resolve more conformational states.
> The authors found that the rmsd values among different protomers chains were relatively small, thus reflecting minimal structural arrangements in the enzymatic core.
>> I thank the authors for the clarifications, and agree that these analyses are indicative of a very rigid and symmetric core structure, and thus justify the application of symmetry.

> The authors attempted methods to parse out more conformational states with variability analysis, but were unsuccessful likely due to the high flexibility of the N-domain.
>> Although unfruitful, I appreciate that the authors tried variability analysis, and believe that this analysis once again highlights the large scale of N-terminal domain movement.

3. Additional validation of subdomain interactions

> The authors performed crosslinking experiments followed by NS-EM to show that the fraction of ALDH1L1 molecules with free, fully mobile N-domains decreases with crosslinking, and added a supplementary figure showing the results.
>> This is a nice addition to support the idea that the N-terminus interacts with the Ct-Int core. Of note, it seems like crosslinking changes the appearance of the originally symmetric 4-lobe Ct-Int core to a larger and more asymmetric shape that might be directly or indirectly related to the packing of the N-terminal domain onto the Int-Ct linker. These crosslinked forms might be interesting targets for future studies.

Structure-function studies were also suggested, but I understand that such mutational analyses might be non-trivial, and are not necessarily essential to arrive at the authors' conclusions.

All minor points were addressed aptly as indicated in the response letter, and additional discussions regarding the distance of domain movement and the significance of the tetrameric state of ALDH1L1 were satisfying. My only issue is with "minor point 4" where I suggested an alternative representation in Fig 5e to highlight the broad interface of the Int-Ct domains. The authors used a sphere representation to highlight residues close to the interface, but because the compared structures are quite divergent it is hard to directly compare the surfaces in the figure, i.e., for some structures, the interaction interface is towards the front, while in other structures much of the interaction area is hidden in the back (as it seems for the ALDH1L1 structure). In this case, I believe my suggestion has

possibly done more harm than good, as the figure gets too busy and does not effectively convey the broad interface. Thus, my recommendation would be to revert to Fig. 5e as in the original version.

Overall, the revised manuscript has been improved by providing additional experiments and explanations, and has thoroughly addressed the comments from all reviewers. I am happy to recommend the manuscript to be published immediately in its current form.

REVIEWERS' COMMENTS:

Reviewer #2 (Remarks to the Author):

In the revised version of their manuscript, Tsybovsky et al. satisfactorily addressed my recommendations. Therefore, in my opinion, this revised version could be accepted for publication.

We are happy that we were able to address the recommendations of the Reviewer in a satisfactory manner.

Reviewer #3 (Remarks to the Author):

I would like to thank the authors for providing a comprehensive response to the questions and comments raised about the original submitted manuscript.

The major suggestions that were made in my initial review and the authors' response were as follows:

1. Clarification of the structural state of the enzyme, i.e., why the authors chose to interpret the enzyme as the "resting state", and if the enzyme used for structure determination has any activity.

> In the Methods section, the authors clarified that the purified ALDH1L1 possesses enzymatic activity, but that "the resting state" without substrate was more amenable to structure determination.

>> The comment has resolved a major curiosity that was unanswered in the original submission.

2. A more detailed analysis of different conformational states among protomers, and application of alternative data processing tools to increase resolution and/or resolve more conformational states.

> The authors found that the rmsd values among different protomers chains were relatively small, thus reflecting minimal structural arrangements in the enzymatic core.

>> I thank the authors for the clarifications, and agree that these analyses are indicative of a very rigid and symmetric core structure, and thus justify the application of symmetry.

> The authors attempted methods to parse out more conformational states with variability analysis, but were unsuccessful likely due to the high flexibility of the N-domain.

>> Although unfruitful, I appreciate that the authors tried variability analysis, and believe that this analysis once again highlights the large scale of N-terminal domain movement.

3. Additional validation of subdomain interactions

> The authors performed crosslinking experiments followed by NS-EM to show that the fraction of ALDH1L1 molecules with free, fully mobile N-domains decreases with crosslinking, and added a supplementary figure showing the results.

>> This is a nice addition to support the idea that the N-terminus interacts with the Ct-Int core. Of note, it seems like crosslinking changes the appearance of the originally symmetric 4-lobe Ct-Int core to a larger and more asymmetric shape that might be directly or indirectly related to the packing of the N-terminal domain onto the Int-Ct linker. These crosslinked forms might be interesting targets for future studies.

Structure-function studies were also suggested, but I understand that such mutational analyses might be non-trivial, and are not necessarily essential to arrive at the authors' conclusions.

All minor points were addressed aptly as indicated in the response letter, and additional discussions regarding the distance of domain movement and the significance of the tetrameric state of ALDH1L1 were satisfying. My only issue is with "minor point 4" where I suggested an alternative representation in Fig 5e to highlight the broad interface of the Int-Ct domains. The authors used a sphere representation to highlight residues close to the interface, but because the compared structures are quite divergent it is hard to directly compare the surfaces in the figure, i.e., for some structures, the interaction interface is towards the front, while in other structures much of the interaction area is hidden in the back (as it seems for the ALDH1L1 structure). In this case, I believe my suggestion has possibly done more harm than good, as the figure gets too busy and does not effectively convey the broad interface. Thus, my recommendation would be to revert to Fig. 5e as in the original version.

Overall, the revised manuscript has been improved by providing additional experiments and explanations, and has thoroughly addressed the comments from all reviewers. I am happy to recommend the manuscript to be published immediately in its current form.

We are delighted to know that the Reviewer is satisfied with our responses. In the revised manuscript, Fig. 5e has been reverted to its original version. We would like to thank the Reviewer for all the thoughtful suggestions.